# Crbn modulates calcium influx by regulating Orai1 during efferocytosis

Hyunji Moon [1,2], Chanhyuk Min[1,2], Gayoung Kim[1], Deokhwan Kim[1,2], Kwanhyeong Kim[1,2], Sang-Ah Lee[1,2], Byeongjin Moon[1,2], Susumin Yang[1,2], Juyeon Lee[1,2], Seung-Joo Yang[1], Steve K. Cho [1], Gwangrog Lee [1,2], Chang Sup Lee[3], Chul-Seung Park [1] & Daeho Park [1,2,4 ✉]

Calcium flux regulating intracellular calcium levels is essential and modulated for efficient efferocytosis. However, the molecular mechanism by which calcium flux is modulated during efferocytosis remains elusive. Here, we report that Orai1, a Crbn substrate, is upregulated via its attenuated interaction with Crbn during efferocytosis, which increases calcium influx into phagocytes and thereby promotes efferocytosis. We found that Crbn deficiency promoted phagocytosis of apoptotic cells, which resulted from facilitated phagocytic cup closure and was nullified by a CRAC channel inhibitor. In addition, Orai1 associated with Crbn, resulting in ubiquitination and proteasomal degradation of Orai1 and alteration of SOCE-mediated calcium influx. The association of Orai1 with Crbn was attenuated during efferocytosis, leading to reduced ubiquitination of Orai1 and consequently upregulation of Orai1 and calcium influx. Collectively, our study reveals a regulatory mechanism by which calcium influx is modulated by a Crbn-Orai1 axis to facilitate efferocytosis.

[1] School of Life Sciences, Gwangju Institute of Science and Technology, Gwangju 61005, Korea. [2] Cell Mechanobiology Laboratory, Gwangju Institute of Science and Technology, Gwangju 61005, Korea. [3] College of Pharmacy and Research Institute of Pharmaceutical Sciences, Gyeongsang National University, Jinju 52828, Korea. [4] Research Center for Cellular Homeostasis, Ewha Womans University, Seoul 03760, Korea. ✉email: daehopark@gist.ac.kr

Efferocytosis is a cellular process that clears apoptotic cells generated during biological processes such as development and tissue homeostasis in multicellular organisms[1]. Studies conducted over the past several decades have revealed the mechanisms of apoptotic cell clearance and determined the consequences of efferocytosis[2–4]. One important feature of efferocytosis is that phagocytes swiftly and continuously remove apoptotic cells. This is achieved by elegant engulfment machinery, which specifically phagocytoses apoptotic cells and is subsequently reprogrammed. Apoptotic cells are recognized by phosphatidylserine (PS)-binding proteins including PS receptors following exposure of PS on the outer leaflet of their plasma membrane[3,5–8]. Thereafter, apoptotic cells are ingested and phagocytes are reprogrammed to continuously remove these cells. The alterations that occur in phagocytes following engulfment of apoptotic cells upregulate genes involved in recognition, internalization, and digestion of apoptotic cells such as Mertk, Drp-1, and Ucp2[9–11].

Calcium, whose intracellular levels are tightly regulated, is essential for prompt and continuous efferocytosis. Specifically, calcium is required for recognition of PS on apoptotic cells by phagocytes, which results in swift and continuous engulfment of apoptotic cells[11–16]. Both extracellular calcium and calcium in intracellular stores are indispensable for efferocytosis. Thus, calcium flux and genes required for calcium flux are crucial for efferocytosis[11,15,16]. One of pivotal mechanisms to maintain the balance of intracellular calcium levels is store-operated calcium entry (SOCE) mediated by calcium release-activated calcium (CRAC) channels called Orais and calcium sensors in the endoplasmic reticulum (ER) called Stims[17–20]. Calcium influx mediated by SOCE is reportedly indispensable for efficient internalization of apoptotic cells and post-engulfment responses such as transforming growth factor-β production[16]. In addition to calcium influx, it was recently reported that Drp-1-mediated mitochondrial fission, which blocks Mcu-mediated mitochondrial calcium sequestration, is required for vesicular trafficking and phagosomal degradation during efferocytosis and thereby facilitates continuous removal of apoptotic cells[11]. However, although several studies have focused on the importance of calcium flux for efferocytosis, the signaling pathway that induces calcium flux and the mechanism by which it is modulated during efferocytosis are incompletely understood.

Cereblon (Crbn), which was originally identified as a gene associated with intellectual disability, is a component of the CRL4$^{CRBN}$ E3 ubiquitin ligase. Crbn functions as a substrate acceptor in this E3 ubiquitin ligase and determines its substrate specificity. Thus, Crbn mediates the ubiquitination and subsequent proteasomal degradation of its substrates[21–24]. Several substrates of Crbn, which include transmembrane and cytosolic proteins, have been identified[24–29]. Ampk, a well-characterized cytosolic substrate of Crbn, is ubiquitinated and degraded through its biochemical interaction with Crbn. Ampk activity is increased in $Crbn^{−/−}$ mice, resulting in resistance to high-fat diet-induced obesity[26,27]. Recent studies have focused on Crbn as a cellular target of immunomodulatory drugs (IMiDs) such as thalidomide. IMiDs bind to a surface groove in the C-terminus of Crbn. This binding alters the substrate specificity of Crbn, resulting in degradation of neo-substrates, such as IKZF1, CK1a, and GSPT1 or decreased degradation of its endogenous substrates such as MEIS2[23,24,30–32]. These alterations are considered to underlie the immunomodulatory effects and teratogenic activity of IMiDs.

The activation of Ampk in Crbn-depleted cells and during efferocytosis led us to question whether Crbn influences the clearance of apoptotic cells[26,27,33–35]. We attempted to answer this question in the current study. We found that Crbn negatively regulated efferocytosis by phagocytes. However, unexpectedly, the effects of Crbn on efferocytosis were unlikely due to Ampk activation or other known Crbn substrates. Using genetic, biochemical, and pharmaceutical approaches, we found that Crbn interacted with Orai1, which mediated SOCE, resulting in ubiquitination and subsequent degradation of Orai1 through the ubiquitin–proteasome pathway. Consequently, Crbn negatively regulated calcium influx. Interestingly, the interaction of Orai1 with Crbn was weakened during efferocytosis, which reduced ubiquitination and consequently increased the level of Orai1. These effects increased calcium influx and the efficiency of efferocytosis. Collectively, this study demonstrates unappreciated machinery that modulates calcium influx to promote removal of apoptotic cells through a Crbn–Orai1 axis in efferocytosis.

## Results

**Crbn negatively regulates engulfment of apoptotic cells.** Previous studies showing that Ampk is activated during efferocytosis and in Crbn-depleted cells led us to investigate whether Crbn affects engulfment of apoptotic cells[27,33,34]. To test the effects of Crbn on efferocytosis, LR73 cells and immortalized mouse embryonic fibroblasts (MEFs) overexpressing Crbn were incubated with TAMRA-stained apoptotic thymocytes, and efferocytosis by Crbn-overexpressing cells was measured using flow cytometry. Interestingly, a lower percentage of Crbn-overexpressing phagocytes than control phagocytes engulfed apoptotic cells. In addition, the number of apoptotic cells ingested per phagocyte was lower for Crbn-overexpressing cells than for control cells, as indicated by the mean fluorescence intensity (MFI, which represented the relative number of apoptotic cells per phagocyte) (Fig. 1a and Supplementary Fig. 1a, b). As an alternative approach to validate the effects of Crbn on efferocytosis, we used MEFs derived from $Crbn^{−/−}$ mice (Supplementary Fig. 2a). In contrast with the effects of Crbn overexpression, efferocytosis by $Crbn^{−/−}$ MEFs was more efficient than that by wild-type (WT) MEFs, as measured by the percentage of phagocytes that engulfed apoptotic cells and the MFI of engulfing phagocytes. Notably, the phenotype of efferocytosis by $Crbn^{−/−}$ MEFs was rescued by Crbn overexpression (Fig. 1b), suggesting that the altered efferocytosis by $Crbn^{−/−}$ MEFs was caused by loss of Crbn. We further tested whether professional phagocytes derived from $Crbn^{−/−}$ mice also exhibit altered engulfment of apoptotic cells (Supplementary Fig. 2b, c). Bone marrow-derived macrophages (BMDMs) and peritoneal macrophages derived from $Crbn^{−/−}$ mice consistently showed a superior ability to phagocytose apoptotic cells than those derived from WT mice (Fig. 1c, d), and the superior ability of efferocytosis by professional phagocytes derived from $Crbn^{−/−}$ mice was not due to the difference of the degree of differentiation (Supplementary Fig. 3a, b). Next, we tested whether Crbn could also affect phagocytosis of non-apoptotic targets, such as polystyrene beads and E. coli and S. aureus particles. Phagocytosis of the particles by $Crbn^{−/−}$ MEFs was comparable with that by WT MEFs (Supplementary Fig. 4a, b). These data indicate that Crbn negatively and specifically regulates phagocytosis of apoptotic cells.

**Clearance of apoptotic cells is augmented in Crbn$^{−/−}$ mice.** Next, we tested whether Crbn deficiency changes the ability of phagocytes to clear apoptotic cells in vivo using two different approaches. First, TAMRA-labeled apoptotic thymocytes were injected into the peritoneum of WT or $Crbn^{−/−}$ mice, and engulfment of these cells by peritoneal macrophages was measured. Apoptotic cells were engulfed by about 43% of $Crbn^{−/−}$ peritoneal macrophages, but only by 28% of WT peritoneal

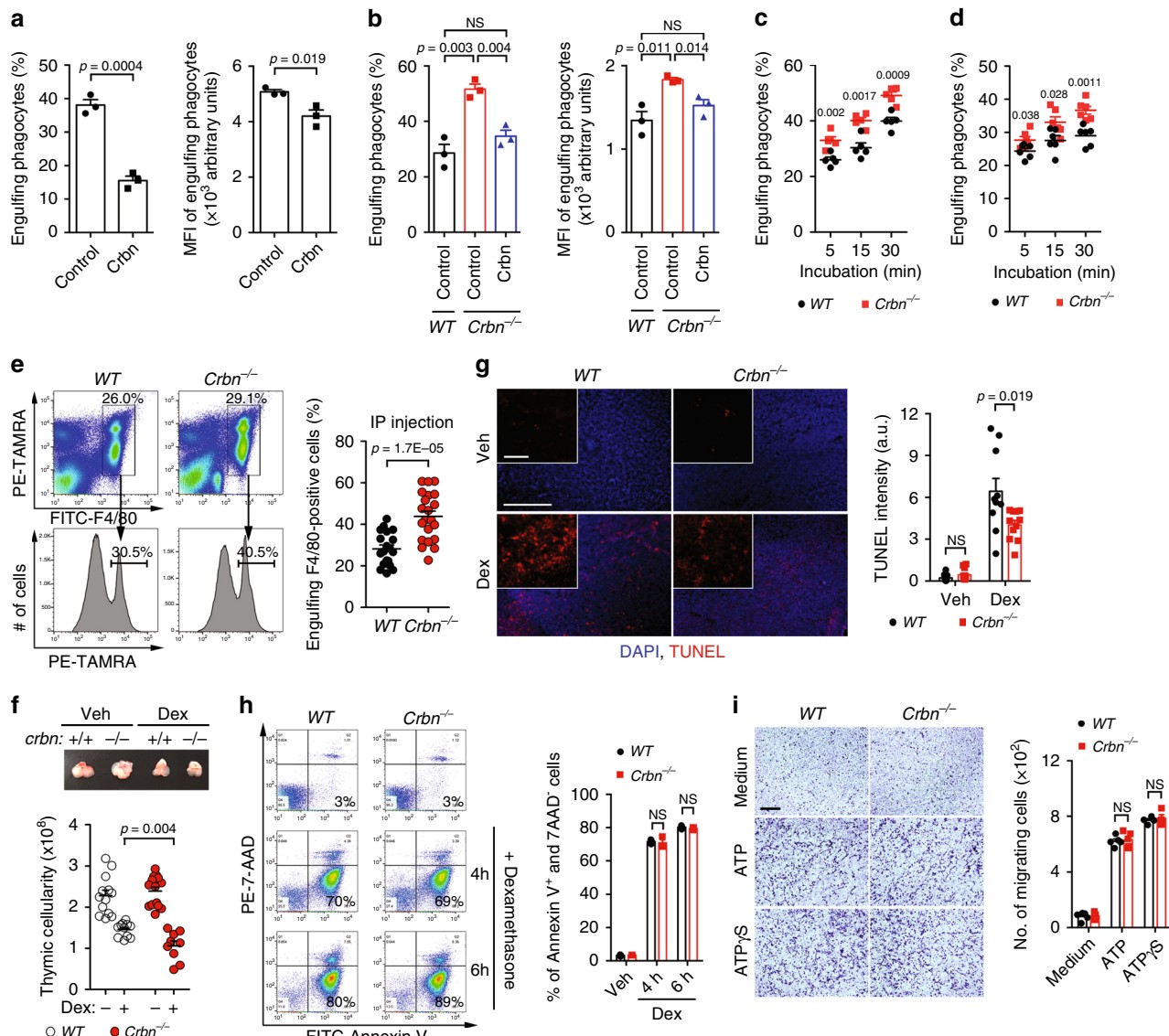

**Fig. 1 Crbn negatively regulates phagocytosis of apoptotic cells. a** LR73 cells transfected with Crbn were incubated with TAMRA-stained apoptotic cells for 2 h and analyzed by flow cytometry. *n* = 3 experiments, mean ± SEM (two-tailed unpaired Student *t* test). **b** MEFs derived from *WT* or *Crbn*⁻/⁻ mice or *Crbn*⁻/⁻ MEFs transfected with Crbn were incubated with TAMRA-stained apoptotic cells for 2 h and analyzed by flow cytometry. *n* = 3 experiments, mean ± SEM. NS, not significant (one-way ANOVA). **c, d** BMDMs (**c**, *n* = 5 mice) or peritoneal macrophages (**d**, *n* = 6 mice) derived from *Crbn*⁻/⁻ or *WT* mice were incubated with TAMRA-stained apoptotic cells for the indicated durations and analyzed by flow cytometry, mean ± SEM. The numbers in the graphs indicate *p* values (two-way ANOVA). **e** TAMRA-stained apoptotic cells were peritoneally injected into *WT* (*n* = 19 mice) or *Crbn*⁻/⁻ (*n* = 21 mice) mice. At 15 min after injection, peritoneal exudates were stained with F4/80 and analyzed by flow cytometry. *n* = 3 experiments, mean ± SEM (two-tailed unpaired Student *t* test). **f** Mice were intraperitoneally injected with 250 µg dexamethasone dissolved in PBS. At 4 h after injection, the sizes of the thymi were observed (top) and the number of thymocytes in the thymi was counted (bottom). *n* = 5 experiments (each dot represents one mouse), mean ± SEM (two-way ANOVA). **g** The thymi of the mice **f** were stained with TUNEL. Microscopy images were acquired (left) and the intensity of TUNEL was measured with ImageJ (right). Scale bar, 250 or 50 µm (inset). *n* = 3 experiments (each dot represents one slide), mean ± SEM (two-tailed unpaired Student *t* test). **h** Thymocytes from the indicated mice were incubated with 50 µM dexamethasone for the indicated duration, stained with Annexin V and 7-AAD, and analyzed by flow cytometry. *n* = 3 experiments, mean ± SEM. NS, not significant (two-tailed unpaired Student *t* test). **i** BMDMs from the indicated mice migrating to 100 nM ATP or ATPγS were measured using the transwell cell migration assay. Scale bar, 200 µm. *n* = 4 experiments, mean ± SEM. NS, not significant (Two-tailed unpaired Student *t* test).

macrophages (Fig. 1e). Second, rapid and synchronous death of thymocytes upon injection of dexamethasone and the subsequent clearance of apoptotic thymocytes by resident phagocytes provides an in vivo model of apoptotic cell clearance[10]. Thus, clearance of apoptotic cells by *Crbn*⁻/⁻ phagocytes was validated using this method. The thymus of *Crbn*⁻/⁻ mice was smaller than that of *WT* mice at 4 h after dexamethasone injection (Fig. 1f, top). Calculation of the absolute number of thymocytes

confirmed that there were fewer thymic cells in *Crbn*⁻/⁻ mice than in *WT* mice after dexamethasone injection (Fig. 1f, bottom). Notably, the decrease in thymic cellularity in *Crbn*⁻/⁻ mice was due to augmented clearance of apoptotic cells but not the altered rate of apoptosis or cell migration. Specifically, the fluorescence intensity of TdT-mediated dUTP nick end labeling (TUNEL) was lower in *Crbn*⁻/⁻ mice than in *WT* mice following dexamethasone injection (Fig. 1g). By contrast, the percentage of

apoptotic thymocytes induced by dexamethasone and the number of migrating macrophages were comparable between cells derived from $Crbn^{-/-}$ and $WT$ mice (Fig. 1h, i). These data indicate that Crbn depletion promotes clearance of apoptotic cells in vivo as well as in vitro.

**Known Crbn substrates do not alter engulfment of apoptotic cells**. Next, we investigated the mechanism by which Crbn could adjust efferocytosis. We initially hypothesized that Crbn regulates efferocytosis through its substrates. Ampk was a potential candidate through which Crbn affected efferocytosis given the association of Ampk activation with efferocytosis and Crbn. To investigate this, we first explored Ampk activation in Crbn-depleted cells. As reported, the level of phosphorylated Ampk was higher in BMDMs derived from $Crbn^{-/-}$ mice than in those derived from $WT$ mice (Supplementary Fig. 5a)[26,27]. We next tested whether Ampk activation facilitates efferocytosis. Although a constitutive active form of Ampk-activated Ampk signaling pathways, which was confirmed by observing the level of phosphorylation of Raptor and S6k (Supplementary Fig. 5b), unexpectedly, it failed to promote efferocytosis (Supplementary Fig. 5c). Moreover, Ampk was not activated in any tested type of phagocytes during efferocytosis in our experimental conditions (Supplementary Fig. 5d), suggesting that Ampk is not the substrate by which Crbn affects efferocytosis.

Thus, we investigated whether other known Crbn substrates, namely, Slo, glutamate synthetase, CIC-1/2, and MEIS2, mediate the effects of Crbn on efferocytosis by cells overexpressing each substrate or pharmacological approaches. The level of efferocytosis by LR73 cells overexpressing Slo or glutamate synthetase was comparable with that by control cells (Supplementary Fig. 6a, b). In addition, treatment of BMDMs with 9-AC, a chloride-channel blocker inhibiting CIC-1/2, or macrophages derived from THP-1 cells with lenalidomide, an IMiD that alters the substrate specificity of Crbn for MEIS2, failed to inhibit or promote efferocytosis (Supplementary Fig. 6c, d). These data suggest that the known substrates of Crbn are not involved in the effects of Crbn on efferocytosis.

**Crbn modulates calcium influx during efferocytosis**. Due to the lack of relevance of the known Crbn substrates to efferocytosis, we monitored which steps of efferocytosis are affected by Crbn. We first tested whether the effects of Crbn on efferocytosis could be caused by the different number of bound apoptotic cells to phagocytes, and if so, whether it could occur in a PS-dependent manner using Cy3-labeled PS beads. A comparable number of apoptotic cells or PS beads bound to $Crbn^{-/-}$ and $WT$ BMDMs upon incubation at 4 °C (Fig. 2a), suggesting that Crbn affects a process following apoptotic cell binding to phagocytes in efferocytosis. Thus, we next tested whether Crbn functions during internalization of targets. To address this, we compared the actin polymerization, a key feature of phagocytes engulfing targets and required for internalization of targets, around PS beads being engulfed by $WT$ or $Crbn^{-/-}$ BMDMs. The levels and intensities of F-actin enrichment around PS beads, indicated by phalloidin, could not be distinguished between $WT$ and $Crbn^{-/-}$ BMDMs (Fig. 2b, Supplementary Fig. 7, and Supplementary Movie 1). However, SiR-actin, a live cell fluorogenic F-actin-labeling probe, and time-lapse confocal microscopy revealed that the duration of F-actin around PS beads decreased in $Crbn^{-/-}$ BMDMs compared with that in $WT$ BMDMs (Fig. 2c, d and Supplementary Movie 2). The disassembly of F-actin is synchronized with phagocytic cup closure in efferocytosis[36]. Thus, we compared phagocytic cup closure between $WT$ and $Crbn^{-/-}$ BMDMs. Interestingly, the phagocytic cups closed earlier in $Crbn^{-/-}$

BMDMs than in $WT$ BMDMs. In addition, the time required for phagocytic cup closure approximately coincided with the duration for F-actin (Fig. 2e, f and Supplementary Movie 3). These data imply that Crbn acts on efferocytosis through affecting phagocytic cup closure.

The intracellular calcium level of phagocytes increases during efferocytosis and calcium signals are often observed at the site of phagocytic cups. In addition, calcium depletion or blockade of calcium flux abrogates phagocytic cup formation[15,16,37–39]. Moreover, calcium entry in CD4+ T cells derived from $Crbn^{-/-}$ mice is increased upon T-cell receptor stimulation[40]. These correlations between the effects of Crbn or calcium on efferocytosis and the previous report led us to investigate whether Crbn could modulate the level of calcium in phagocytes engulfing apoptotic cells. To this end, BMDMs stained with Fluo3-AM were incubated with apoptotic thymocytes, and the intracellular calcium level, represented by Fluo3 fluorescence, was monitored. As reported previously, the intracellular calcium level continuously increased in both $Crbn^{-/-}$ and $WT$ BMDMs upon addition of apoptotic cells[11,15,16]. Intriguingly, however, stimulation with apoptotic cells increased the intracellular calcium level more rapidly and to a greater extent in $Crbn^{-/-}$ BMDMs than in $WT$ BMDMs (Fig. 2g), which was also observed using Fura2-AM, a fluorescent ratiometric indicator of intracellular calcium (Supplementary Fig. 8). The higher calcium level in $Crbn^{-/-}$ BMDMs was likely due to entry of extracellular calcium, not to release of calcium from intracellular stores, because calcium flux in $Crbn^{-/-}$ BMDMs stimulated with apoptotic cells was indistinguishable from that in $WT$ BMDMs in the absence of extracellular calcium (Fig. 2h), which implies that Crbn affects calcium influx into phagocytes during efferocytosis.

**SOCE-mediated calcium influx is increased in $Crbn^{-/-}$ phagocytes**. SOCE is a central mechanism that induces calcium influx, and CRAC channels mediating SOCE are required for efferocytosis[15–17]; therefore, we next tested whether Crbn modulates calcium influx through intrinsic regulation of SOCE. To this end, calcium in the ER was depleted by thapsigargin and calcium entry was monitored using Fura2-AM. SOCE, represented by the peak F340/380, was 13% higher in BMDMs derived from $Crbn^{-/-}$ mice than in those derived from $WT$ mice, and the rate of calcium influx, as indicated by the slope (360s–444s), was also significantly higher in the former cells than in the latter cells. Notably, AUC, indicating calcium release from intracellular stores, did not differ between $Crbn^{-/-}$ and $WT$ BMDMs (Fig. 3a), again suggesting that calcium release from intracellular stores does not contribute to the increased calcium flux observed in $Crbn^{-/-}$ BMDMs. SOCE was also higher in peritoneal macrophages and MEFs derived from $Crbn^{-/-}$ mice than in those derived from $WT$ mice (Supplementary Fig. 9a, b). By contrast, Crbn overexpression elicited the opposite effect on SOCE. Specifically, SOCE was reduced by 12% and by 10% in Crbn-overexpressing LR73 cells and MEFs, respectively, and the rate of calcium entry was also decreased in these cells (Fig. 3b and Supplementary Fig. 10a, b), suggesting that Crbn negatively regulates SOCE.

Next, we investigated whether CRAC channels mediate the alteration of SOCE by Crbn using YM-58483, a CRAC channel inhibitor. Notably, SOCE was comparable in $Crbn^{-/-}$ and $WT$ BMDMs upon treatment with YM-58483 (Fig. 3c), suggesting that Crbn modulates SOCE mediated by CRAC channels. Furthermore, we validated the involvement of CRAC channel-mediated SOCE in the effects of Crbn on efferocytosis. EGTA, an extracellular calcium chelator, and YM-58483 completely and similarly abrogated efferocytosis by $Crbn^{-/-}$ and $WT$ BMDMs

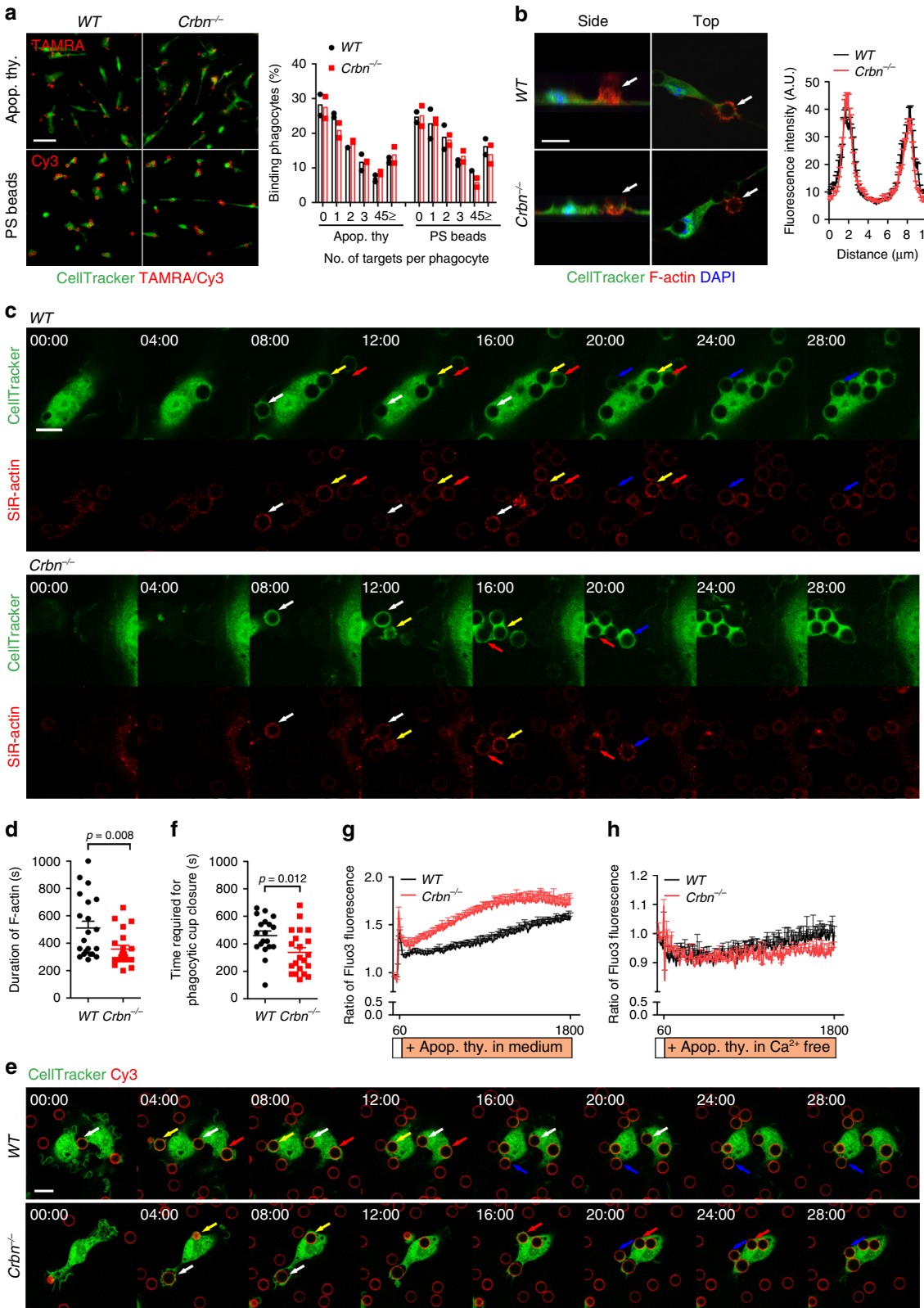

(Fig. 3d, e) and specifically, the time required for phagocytic cup closure during efferocytosis by *Crbn*$^{-/-}$ and *WT* BMDMs were delayed and comparable in the presence of YM-58483 (Fig. 3f, g and Supplementary Movies 4, 5). In addition, another CRAC channel inhibitor, GSK-5498A, similarly inhibited efferocytosis by *Crbn*$^{-/-}$ and *WT* BMDMs (Supplementary Fig. 10c). However, Mdivi-1, a specific inhibitor for Drp-1 which increases the intracellular calcium level by blocking Mcu-mediated mitochondrial calcium sequestration during efferocytosis, inhibited efferocytosis by *WT* BMDMs, but not by *Crbn*$^{-/-}$ BMDMs (Fig. 3h), suggesting that Drp-1-mediated calcium flux is not related to the effects of Crbn on efferocytosis. Taken together, these data imply that Crbn affects efferocytosis by modulating CRAC channel-mediated SOCE.

**Fig. 2 Crbn deficiency promotes phagocytic cup closure. a** The indicated BMDMs were incubated with TAMRA-stained apoptotic cells (top) or Cy3-labeled PS beads (bottom) at 4 °C for 30 min and then extensively washed with PBS to remove unbound targets. Bound targets were observed by microscopy (left). The number of targets per phagocyte were quantified from 50 randomly acquired images (right). Scale bar, 50 μm. $n = 2$ experiments. **b** BMDMs derived from *WT* or *Crbn*$^{-/-}$ mice were incubated with PS beads, stained with phalloidin, and observed by confocal microscopy (left). The intensity of phalloidin across the targets was quantified using ImageJ. Distance 5 μm is the center of the 6 μm PS bead (right). Arrows indicate a PS bead being engulfed. Scale bar, 10 μm. $n = 3$ experiments, mean ± SEM. **c, d** *WT* or *Crbn*$^{-/-}$ BMDMs stained with CellTracker and SiR-actin were incubated with PS beads and the BMDMs phagocytosing the beads were observed by time-lapse confocal microscopy **c**. The duration of F-actin was measured **d**. Arrows with a same color indicate a same PS bead being engulfed. Scale bar, 10 μm. $n = 3$ experiments (each dot represents one PS bead), mean ± SEM (two-tailed unpaired Student $t$ test). **e, f** The indicated BMDMs were incubated with Cy3-labeled PS beads and observed using time-lapse confocal microscopy **e**. Time required for phagocytic cup closure was measured (**f**, $n = 3$ experiments, each dot represents one PS bead, mean ± SEM, two-tailed unpaired Student $t$ test). Arrows with a same color indicate a same PS bead being engulfed. Scale bar, 10 μm. **g, h** BMDMs from the indicated mice were stained with Fluo3-AM, and then apoptotic cells in RPMI containing (**g**, $n = 4$ experiments, mean ± SEM) or lacking (**h**, $n = 3$ experiments, mean ± SEM) calcium were added at the indicated time. Fluorescence of BMDMs was measured with a microplate reader (FlexStation 3).

**Orai1 associates with Crbn**. Next, we investigated how Crbn modulates SOCE mediated by CRAC channels. CRAC channels and calcium sensors in the ER mediate SOCE; therefore, we hypothesized that Orai1, a CRAC channel, and Stim1, a calcium sensor in the ER, may be substrates of Crbn. To test this possibility, the levels of Orai1 and Stim1 were compared between *Crbn*$^{-/-}$ and *WT* phagocytes. Interestingly, the level of Orai1 was much higher in *Crbn*$^{-/-}$ BMDMs, peritoneal macrophages, and MEFs than in their *WT* counterparts, whereas the level of Stim1 was similar in *Crbn*$^{-/-}$ and *WT* phagocytes (Fig. 4a and Supplementary Fig. 11a). In addition, the level of Orai1 was higher in Crbn-depleted 293T cells using CRISPR/Cas9 than in control cells (Fig. 4b and Supplementary Fig. 11b). The elevation of Orai1 in *Crbn*$^{-/-}$ BMDMs was not due to transcriptional regulation because the transcript level of *Orai1* was comparable in *Crbn*$^{-/-}$ and *WT* BMDMs (Fig. 4c). Moreover, in contrast with Crbn depletion, ectopic overexpression of Crbn markedly diminished the level of Orai1 (Fig. 4d), which strongly implies that Crbn is involved in regulating the level of Orai1.

To test whether Orai1 might be regulated as a Crbn substrate, their interaction was first validated using an immunoprecipitation assay. Crbn was robustly co-precipitated with Orai1 both following overexpression and endogenously (Fig. 4e, f and Supplementary Fig. 11c). This association was mediated by the N-terminal cytoplasmic tail of Orai1 and the C-terminus of the Lon domain of Crbn (Supplementary Fig. 12a and Fig. 4g–i). The interaction between Crbn and the N-terminus of Orai1 was further confirmed in yeast where a Crbn homolog is not found (Fig. 4j). These data indicate that Orai1 physically associates with Crbn and could be a substrate of Crbn.

**Orai1 is ubiquitinated and degraded in a Crbn-dependent manner**. Next, to determine whether Orai1 is a substrate of the CRL4$^{Crbn}$ E3 ligase and its level is regulated through the ubiquitin–proteasome pathway, we first explored whether Orai1 forms a complex with the ubiquitin ligase. Crbn, DDB1, and Cul4A, which are the components of the CRL4$^{Crbn}$ E3 ligase, were co-precipitated with Orai1 (Fig. 5a), and the Crbn truncation mutants interacting with Orai1 could also precipitate DDB1 and Cul4A (Supplementary Fig. 12b), implying that Orai1 forms a complex with this E3 ligase. Next, Orai1 ubiquitination was measured following expression of HA-ubiquitin and Orai1. Orai1 was noticeably ubiquitinated (Fig. 5b); however, this was drastically diminished when Crbn was depleted using siRNA (Fig. 5c) and also substantially decreased when lysine residue 89 of Orai1 was mutated (Fig. 5d). We then investigated whether Orai1 is degraded by the proteasome using MG132, a specific proteasome inhibitor. Treatment with MG132 increased the level of Orai1 in *WT* BMDMs, but did not alter that in *Crbn*$^{-/-}$ BMDMs (Fig. 5e). These data indicate that Orai1 is a Crbn substrate whose level is regulated through the ubiquitin–proteasome pathway.

**Orai1 is upregulated by attenuation of its association with Crbn during efferocytosis**. Next, to confirm that Crbn modulates efferocytosis by regulating the level of Orai1, we investigated whether Orai1 expression rescues the decrease in efferocytosis induced by Crbn expression. Engulfment of apoptotic cells by LR73 cells was consistently diminished by Crbn overexpression, but promoted by Orai1 overexpression. However, the inhibitory effect of Crbn on efferocytosis was nullified upon co-expression of Orai1 (Fig. 6a). To further confirm that Crbn-modulated efferocytosis is caused by regulating the level of Orai1, we evaluated the effect of Orai1$^{K89A}$, a mutant with substantially reduced ubiquitination, on efferocytosis. Efferocytosis by phagocytes expressing Orai1$^{K89A}$ was similar to that by phagocytes expressing Orai1. However, efferocytosis mediated by Orai1$^{K89A}$ was less efficiently suppressed by Crbn co-expression compared to efferocytosis mediated by Orai1: a higher percentage of phagocytes expressing Orai1$^{K89A}$ and Crbn than phagocytes expressing Orai1 and Crbn engulfed apoptotic cells (Fig. 6b). These were linked to the levels of Orai1$^{K89A}$ and Orai1 in phagocytes. There was no difference between the level of Orai1$^{K89A}$ and Orai1 when they were expressed alone, but the level of Orai1$^{K89A}$ was higher than that of Orai1 when Crbn was co-expressed (Fig. 6c and Supplementary Fig. 13). These data suggest that Crbn affects efferocytosis through regulating the level of Orai1.

Next, we investigated the relevance of Crbn-mediated Orai1 regulation during efferocytosis. Intriguingly, the level of Orai1 was increased when phagocytes were incubated with apoptotic cells (Fig. 6d and Supplementary Fig. 14a). Notably, this increase of Orai1 was not due to Orai1 derived from apoptotic thymocytes because Orai1 in apoptotic thymocytes was undetectable (Supplementary Fig. 14b). In contrast to the increase of Orai1, ubiquitination of Orai1 was inversely decreased (Fig. 6e). These results might have been due to downregulation of Crbn during engulfment of apoptotic cells; however, this was not the case. Stimulation with apoptotic cells did not affect the Crbn level in phagocytes (Fig. 6f). An alternative explanation for the upregulation of Orai1 upon stimulation with apoptotic cells is that the interaction of Orai1 with Crbn is disrupted, thereby reducing ubiquitination and degradation of Orai1 during efferocytosis. To investigate this, phagocytes were stimulated with apoptotic cells and then the interaction between Crbn and Orai1 was evaluated. Remarkably, the association of Orai1 with Crbn was weakened when phagocytes were incubated with apoptotic cells (Fig. 6g), suggesting that attenuation of the interaction between Crbn and Orai1 during efferocytosis reduces ubiquitination and degradation of Orai1 and thereby increases the level of Orai1. Next, we investigated a mechanism by which the interaction between Crbn and Orai1 is attenuated during efferocytosis. It is possible that Crbn is modified (e.g., phosphorylation) and/or its subcellular localization is altered upon apoptotic cell stimulation, which disrupts the interaction between these two proteins. However,

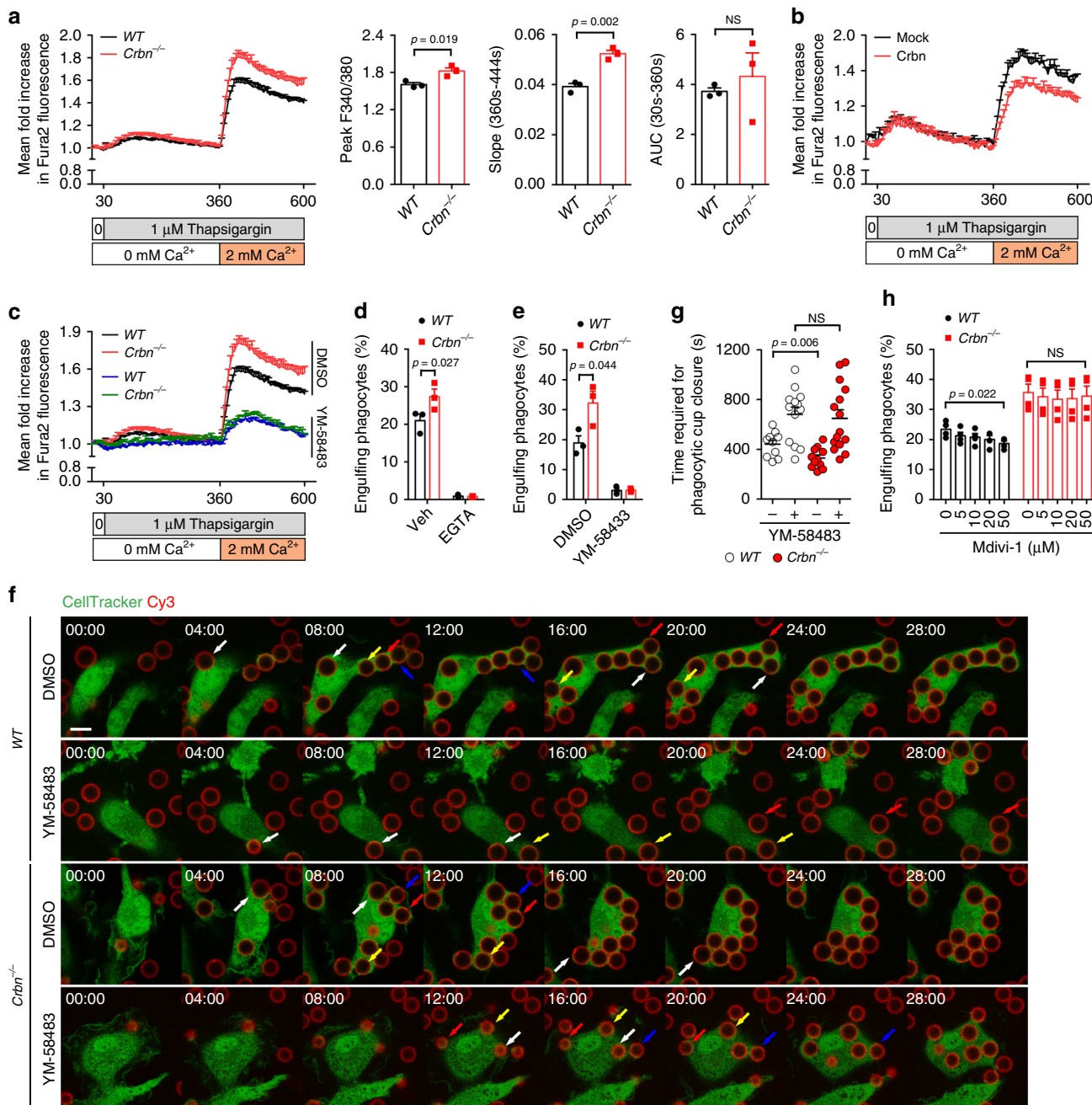

**Fig. 3 Crbn modulates SOCE mediated by CRAC channels. a, b** BMDMs from the indicated mice **a** or Crbn-overexpressing LR73 cells **b** were stained with Fura2-AM and then treated with 1 μM thapsigargin for the indicated duration. Thereafter, 2 mM calcium was added to the cells at the indicated time. Fluorescence of the cells was measured with a microplate reader. AUC area under curve. $n = 3$ experiments, mean ± SEM. NS not significant (two-tailed unpaired Student $t$ test). **c** BMDMs derived from *WT* or *Crbn*$^{-/-}$ mice were pre-incubated with 10 μM YM-58483 and then SOCE was measured as in **a**. **d, e** BMDMs derived from *WT* or *Crbn*$^{-/-}$ mice were incubated with TAMRA-stained apoptotic cells in 5 mM EGTA **d** or 10 μM YM-58483 **e**. Thereafter, engulfing phagocytes were analyzed by flow cytometry. $n = 3$ experiments, mean ± SEM (two-tailed unpaired Student $t$ test). **f, g** BMDMs derived from *WT* and *Crbn*$^{-/-}$ mice were incubated with Cy3-labeled PS beads and observed using time-lapse confocal microscopy **f**. Time required for phagocytic cup closure was measured **g**. Arrows with a same color indicate a same PS bead being engulfed. Scale bar, 10 μm. $n = 3$ experiments (each dot represents one PS bead), mean ± SEM. NS not significant (two-way ANOVA). **h** BMDMs from indicated mice were incubated with TAMRA-stained apoptotic cells in the indicated concentration of Mdivi-1, and engulfing phagocytes were analyzed by flow cytometry. $n = 4$ experiments, mean ± SEM. NS not significant (one-way ANOVA).

neither the subcellular localization nor the tyrosine-phosphorylation of Crbn was affected upon addition of apoptotic cell (Fig. 6h, i and Supplementary Fig. 15). Alternatively, the Orai1–Stim1 interaction, which occurs during SOCE[41,42], may disrupt the basal Crbn–Orai1 interaction. Stim1, a calcium sensor on the ER membrane, transiently interacts with Orai1 and thereby

induces entry of extracellular calcium through Orai1 specifically when the ER calcium level decreases. Thus, Stim1 may compete with Crbn for binding to Orai1 if efferocytosis provokes interaction between Orai1 and Stim1. To validate this, we tested whether interaction between Orai1 and Stim1 is induced upon apoptotic cell stimulation. Interaction between Orai1 and Stim1

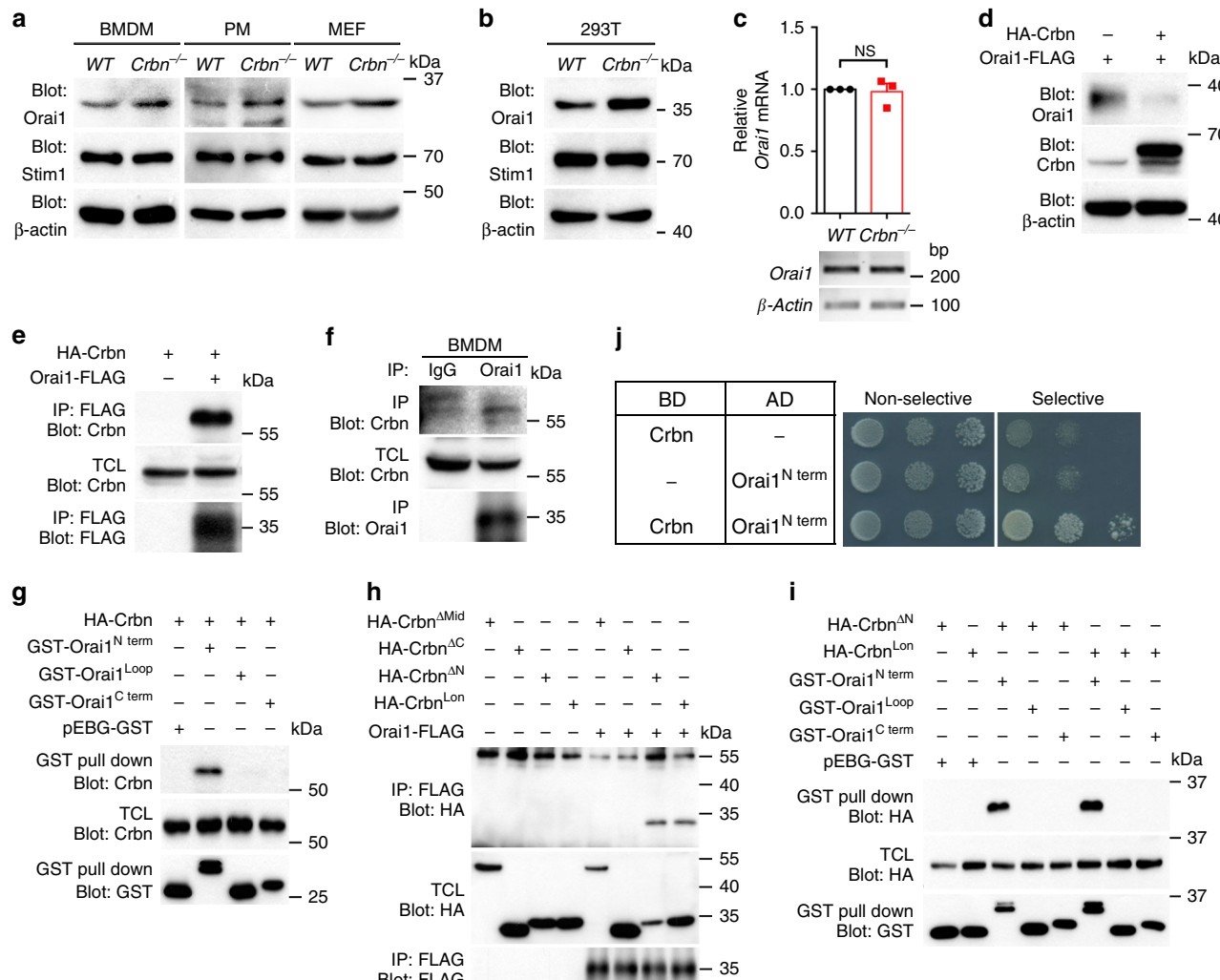

**Fig. 4 Crbn interacts with Orai1. a, b** Phagocytes from the indicated mice **a** or *Crbn* deleted 293T cells by CRISPR/Cas9 **b** were lysed and proteins in the lysates were detected using immunoblotting. Data are representative of three independent experiments. **c** The transcript levels of *Orai1* were analyzed by conventional PCR (bottom) or qRT-PCR (top) using cDNA synthesized from total mRNA extracted from the indicated BMDMs. $n = 3$ experiments, mean ± SEM. NS not significant (two-tailed unpaired Student *t* test). **d** LR73 cells were transfected with the indicated plasmid. At 1 day after transfection, the cells were lysed and proteins in the lysates were detected with the indicated antibodies. Data are representative of three independent experiments. **e, f** 293T cells transfected with the indicated plasmids **e** or BMDMs **f** were lysed and then the lysates were incubated with anti-FLAG antibody-conjugated agarose beads **e** or an anti-Orai1 antibody and protein A/G agarose beads **f**. Bead-bound proteins were detected with the indicated antibodies. Data are representative of five **e** or three **f** independent experiments. IP immunoprecipitation, TCL total cell lysates. **g–i** 293T cells were transfected with the indicated plasmids. At 2 days after transfection, the cells were lysed and the lysates were incubated with agarose beads conjugated with glutathione **g, i** or an anti-FLAG antibody **h**. Bead-bound proteins were detected with the indicated antibodies. Data are representative of four **g** or three **h, i** independent experiments. **j** Yeast cells transformed with the indicated plasmids were plated on selective or non-selective media. Cells on the non-selective media indicate the number of cells plated. BD binding domain, AD activation domain.

was strengthened in phagocytes incubating with apoptotic cells compared with that in control cells (Fig. 6j). In addition, the interaction of Orai1 with Crbn was weakened in cells co-expressing Stim1 with Orai1 and Crbn (Fig. 6k), suggesting that the competitive binding of Stim1 to Orai1 during efferocytosis causes attenuated association of Crbn with Orai1.

Collectively, the data presented in this study suggest that Orai1, which is ubiquitinated and degraded in a Crbn-dependent manner, is upregulated through attenuation of its interaction with Crbn during efferocytosis, which increases calcium influx into phagocytes and thereby facilitates efficient efferocytosis.

## Discussion

This study was prompted by the correlation of Ampk activation with Crbn and efferocytosis. Unexpectedly, Ampk was not

activated during efferocytosis and Ampk activation did not promote uptake of apoptotic cells. Indeed, the relevance of Ampk activation during efferocytosis is controversial. Some studies reported that stimulation with apoptotic cells does not change the level of Ampk activation and that AICAR, an Ampk activator, promotes efferocytosis through a signaling pathway independent of Ampk[10,43]. The disparity between studies may be due to the use of different experimental conditions, e.g., type of phagocytes. We have used various cell types, incubation durations, and numbers of apoptotic cells. However, we could not observe any correlation between efferocytosis and Ampk activation in our experimental conditions. The relationship between Ampk activation and efferocytosis could be further addressed by other investigators.

There are two homologs of Orai1, Orai2 and Orai3. They are all involved in SOCE, although Orai1 plays a major role and has

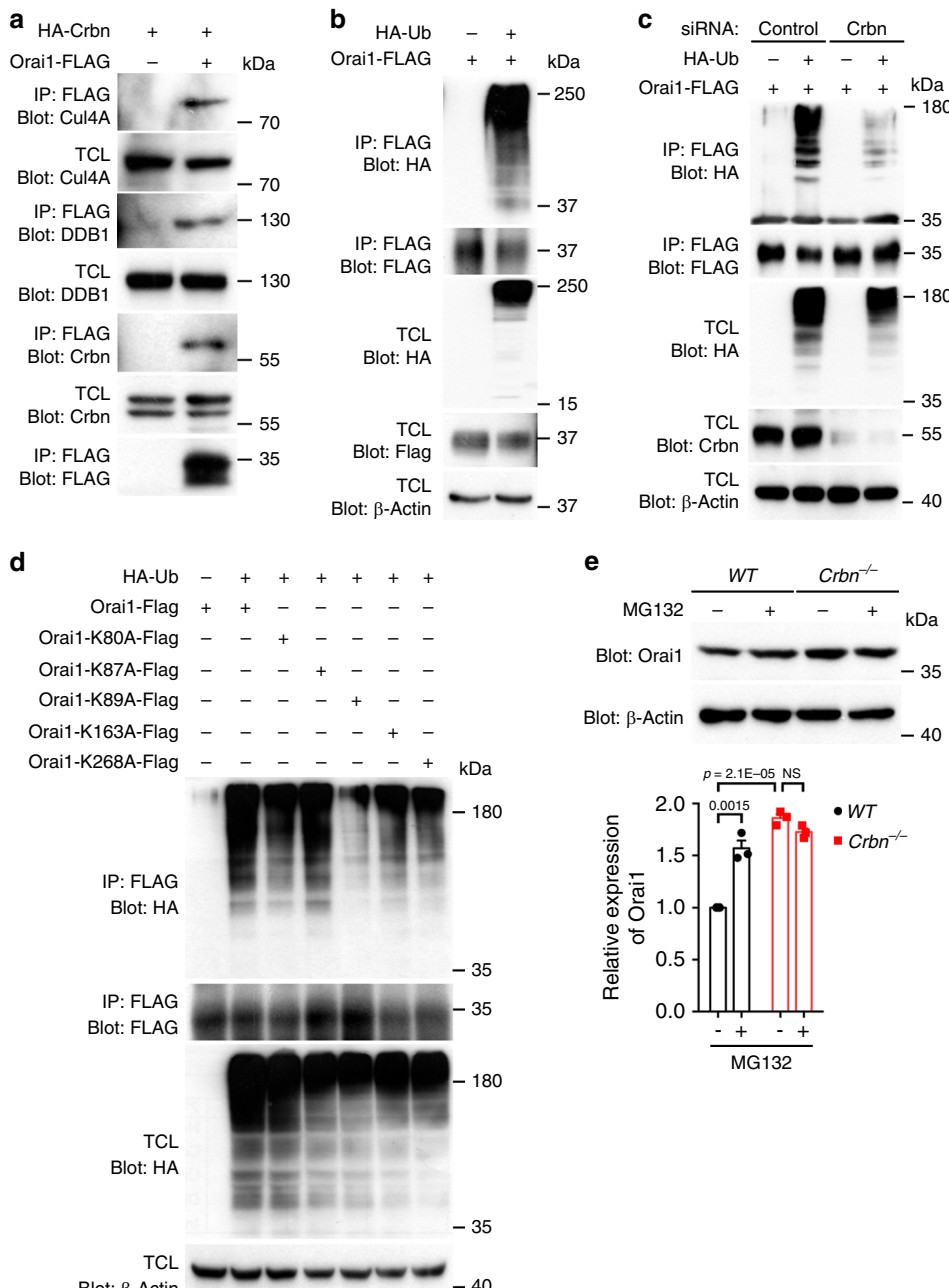

**Fig. 5 Orai1 is ubiquitinated and degraded in a Crbn-dependent manner. a–d** The lysates of 293T cells transfected with the indicated plasmids were incubated with anti-FLAG antibody-conjugated agarose beads. Bead-bound proteins were detected with the indicated antibodies. Data are representative of three **a**, **c**, **d** or four **b** independent experiments. **e** BMDMs derived from the indicated mice were treated with MG132 (5 μM) for 6 h and then lysed. Proteins in the lysates were detected with the indicated antibodies (top) and the levels of Orai1 were quantified (bottom). $n = 3$ experiments, mean ± SEM. The numbers in the graph indicate $p$ values. NS not significant (two-way ANOVA).

been extensively studied[44–46]. We focused on Orai1 in this study; however, Crbn also likely regulates Orai2 and Orai3 in a similar manner to Orai1. Indeed, the level of Orai2 was higher in $Crbn^{-/-}$ phagocytes than in $WT$ phagocytes (Supplementary Fig. 16a). In addition, both Orai2 and Orai3 interacted with Crbn at overexpressing conditions (Supplementary Fig. 16b, c). However, it seems that the effect of Orai2 on efferocytosis was different from that of Orai1, which might result from the discrepant effects on SOCE between Orai1 and Orai2. Although ectopic expression of Orai2 in phagocytes moderately promoted ingestion of apoptotic cells, similar to Orai1 (Supplementary Fig. 16d), only efferocytosis by Orai1-knockdowned MEFs was diminished but

not by Orai2-knockdown MEFs (Supplementary Fig. 16e, f). In addition, SOCE reduction was only observed in Orai1-knockdown MEFs but not in Orai2-knockdowned MEFs (Supplementary Fig. 16g). It is known that SOCE increases in $Orai2^{-/-}$ cells while it decreases in $Orai1^{-/-}$ cells[17,47]. Thus, incomplete depletion of Orai2 and increase of SOCE in $Orai2^{-/-}$ cells may explain the efferocytosis by Orai2-knockdowned MEFs.

It is reportedly that human $ORAI1$ is alternatively translated at residue M64 into the short form of ORAI1, ORAI1β[48–50]. Due to the interaction of Crbn with the N-terminus of Orai1 and the absence of residues 1–63 of ORAI1 in ORAI1β, we tested whether interaction and ubiquitination of ORAI1β differ from those of

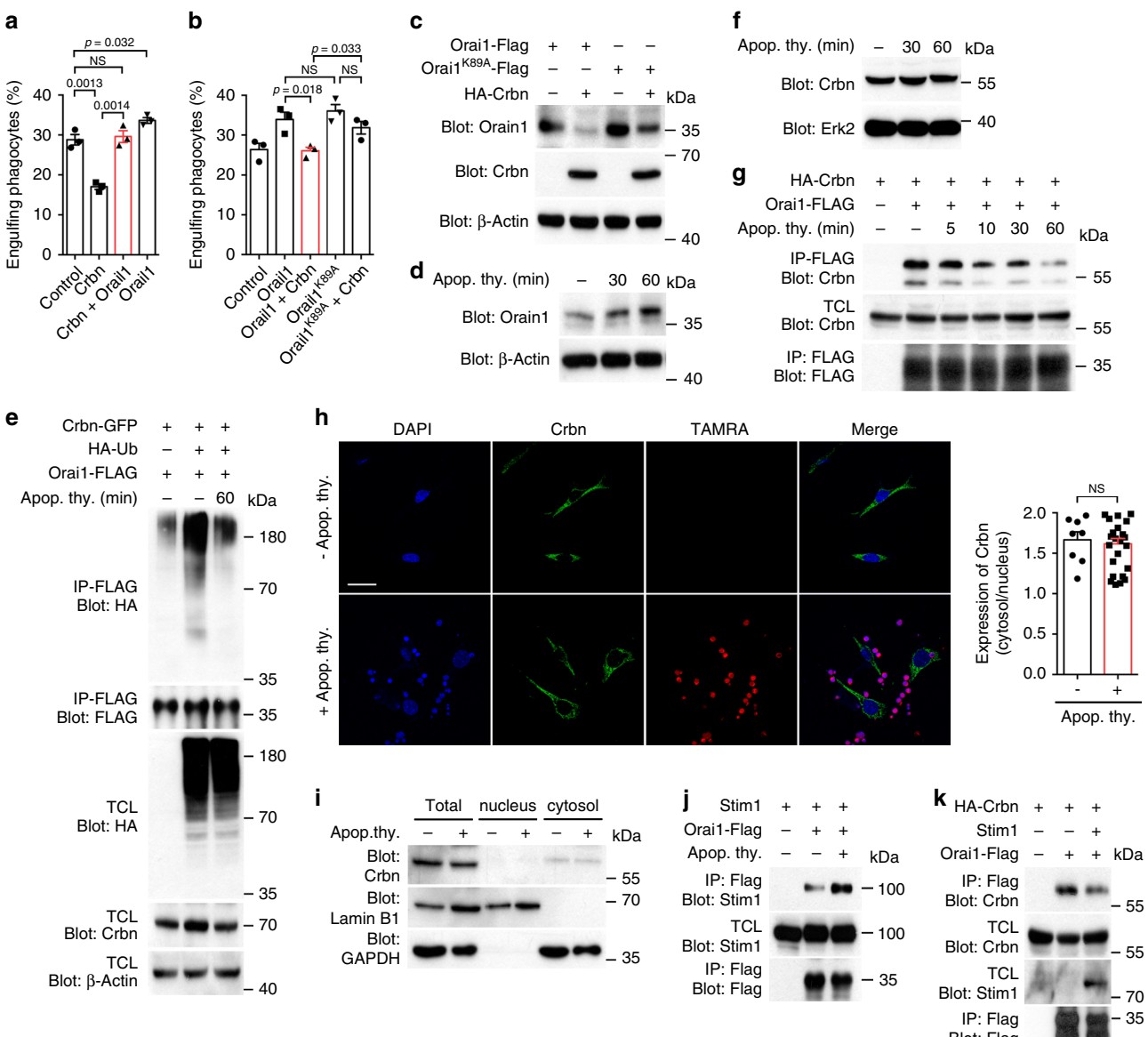

**Fig. 6 The interaction of Orai1 with Crbn is attenuated during efferocytosis. a** LR73 cells transfected with the indicated plasmids were incubated with TAMRA-stained apoptotic cells for 2 h and analyzed by flow cytometry. $n = 3$ experiments, mean ± SEM. The numbers in the graph indicate $p$ values. NS not significant (one-way ANOVA). **b** The experiment was performed as in **a**. $n = 3$ experiments, mean ± SEM. NS not significant (two-tailed unpaired Student $t$ test). **c** LR73 cells transfected with the indicated plasmids were lysed, and the proteins in the lysates were detected by immunoblotting. **d** BMDMs were incubated with apoptotic cells for the indicated durations were lysed, and the levels of Orai1 were detected with an anti-Orai1 antibody. **e** LR73 cells transfected with the indicated plasmids were incubated with apoptotic thymocytes for 60 min, and Orai1 ubiquitination were measured. **f** LR73 cells were incubated with apoptotic thymocytes for the indicated durations and the levels of Crbn were measured. **g** LR73 cells transfected with the indicated plasmids were incubated with apoptotic thymocytes for the indicated durations, and association between Crbn and Orai1 was measured. **h** LR73 cells stimulated with TAMRA-stained apoptotic cells were stained with an anti-Crbn antibody and observed using confocal microscopy (left) and Crbn localization was quantified (right). Scale bar, 20 μm. $n = 8$ cells for control and $n = 25$ cells for adding apoptotic cells, mean ± SEM. NS not significant (two-tailed unpaired Student $t$ test). **i** LR73 cells incubated with apoptotic cells were lysed and fractionized. Proteins in the fractions were detected with the indicated antibodies. **j** LR73 cells transfected with the indicated plasmids were incubated with apoptotic cells, and association between Orai1 and Stim1 was measured. **k** 293T cells were transfected with the indicated plasmids, and association between Crbn and Oria1 was measured in the presence or absence of Stim1. Data are representative of three **c**–**g**, **l**, **k** or five **j** independent experiments. .

ORAI1. Interaction of Crbn with ORAI1β was drastically decreased (Supplementary Fig. 17a), supporting that the N-terminus of ORAI1 is important for the interaction between the proteins. However, unexpectedly, ORAI1β was still ubiquitinated and the ubiquitination level of ORAI1β was comparable with that of ORAI1 (Supplementary Fig. 17b). This may result from the residual interaction of Crbn with ORAI1β which might be enough to ubiquitinate ORAI1β. Another possibility is that

ORAI1β is ubiquitinated by an E3 ligase rather than the CRL4$^{CRBN}$ E3 ligase. The latter is more plausible than the former because indeed ORAI1β ubiquitination was independent from Crbn (Supplementary Fig. 17c) and calcium-dependent inactivation (CDI) of ORAI1β is slower than that of ORAI1, which results in different calcium signaling by ORAI1β from that by ORAI1[50,51]. Thus, a regulatory mechanism by which the level of ORAI1β is modulated may be independent of that of ORAI1.

Slo, a BK channel subunit, was the first Crbn substrate to be identified[25], and several Crbn-interacting proteins were subsequently reported. Interestingly, among them, Slo (potassium channel) and CIC-1/2 (chloride channels) are transmembrane proteins, specifically ion channels. Although Crbn interacts with and mediates the ubiquitination of both these proteins, they are differently regulated. The total level of Slo is not affected by Crbn, but its surface expression level is altered through Crbn-dependent ubiquitination[52]. By contrast, ubiquitination of CIC channels by Crbn induces its degradation, which decreases their total levels[29]. Ubiquitination mediated by Crbn has similar consequences on Orai1 and CIC channels. Similarly, ubiquitination of Orai1 by Crbn induces its degradation, rather than altering its subcellular localization. It will be intriguing to study mechanisms by which Crbn-mediated ubiquitination can be switched between inducing degradation and altering the subcellular localization of a substrate.

Calcium influx occurs very promptly after incubation of phagocytes with apoptotic cells. In addition, phagocytic cup formation requiring calcium influx is instant as well; however, the half-life of Orai1 is about 12 h in 293T cells (Supplementary Fig. 18a) and thus Orai1 is not upregulated as fast. This demonstrates that upregulated Orai1 functions during subsequent engulfment of apoptotic cells by phagocytes, rather than when phagocytes initially encounter apoptotic cells. This is a well-established notion. For example, PPAR-δ, Ucp-2, or Drp-1 confers the ability to phagocytes to continuously remove apoptotic cells. Similarly, Orai1 upregulation could make phagocytes suitable for continuous efferocytosis. Indeed, $Crbn^{-/-}$ BMDMs more rapidly increased the MFI of engulfing phagocytes than WT BMDMs (Supplementary Fig. 18b), indicating that $Crbn^{-/-}$ phagocytes are more capable of continuously engulfing apoptotic cells.

Collectively, Orai1, a Crbn-interacting protein, is ubiquitinated and degraded in a Crbn-dependent manner and is upregulated through its attenuated interaction with Crbn during efferocytosis. This mechanism increases calcium influx into phagocytes and facilitates efficient clearance of apoptotic cells by phagocytes. Crbn is a druggable target, and defects in clearance of apoptotic cells are related to autoimmune-related diseases. Our findings may provide an approach to develop a drug that can be used to treat diseases caused by defects in engulfment of apoptotic cells.

## Methods

**Cell culture and transfection**. 293T cells and MEFs were maintained in DMEM medium supplemented with 10% fetal bovine serum (FBS) and 1% penicillin–streptomycin–glutamine (PSQ). LR73 cells were maintained in alpha-MEM supplemented with 10% FBS and 1% PSQ. Peritoneal exudates were collected after injection of cold PBS into peritoneum and plated on non-culture dishes. 6 h after plating, floating cells were removed and attached cells were used as resident peritoneal macrophages. Peritoneal macrophages were maintained in RPMI medium supplemented with 10% FBS and 1% PSQ. Bone marrow cells were differentiated into BMDMs using RPMI medium containing 20% L929-conditioned medium, 10% FBS, and 1% PSQ. 293T cells, MEFs, and LR73 cells were transfected using Profection mammalian transfection system (Promega), Fugene HD (Promega), and Lipofectamin 2000 (Invitrogen), respectively, according to the manufacturer's instructions. 293T cells were transfected with *control* siRNA (Dharmacon, ON-TARGETplus *control* siRNA, D-001810-01-20) or *crbn* siRNA (Dharmacon, ON-TARGETplus Human *CRBN* siRNA, L-021086-00-0005) using lipofectamine 2000 (Invitrogen). MEFs were transfected with *Orai1* siRNA (Dharmacon, ON-TARGETplus Mouse *Orai1* siRNA, L-056431-02-0005) or *Orai2* siRNA (Dharmacon, ON-TARGETplus Mouse *Orai2* siRNA, L-057985-01-0005) using Fugene HD transfection reagent (Promega, E2311).

**Plasmids and antibodies**. All plasmids used in this study were sequenced to confirm their identity. Crbn, Ampk, Slo, or HA-Ubiquitin constructs used in this study were described previously[26]. Orai1 (MMM1013-202764440), Orai2 (MMM1013-202859855), and Orai3 (MMM1013-202842392) cDNA were purchased from Open Biosystems to construct expression vectors by a PCR-based strategy. Glutamine synthetase (GS) cDNA were synthesized from mRNA of the testis of mice and GS expression vectors were generated. The complete list of all primer sequences used in the study is provided as a Supplementary table. Anti-Flag (Sigma, F1804), anti-HA (Santa Cruz, sc-7392), anti-GST (Santa Cruz, sc-138),

anti-Crbn (Sigma, HPA045910), anti-Orai1 (Santa Cruz, sc-68895), anti-Orai1 (Alomone labs, ACC-062), anti-Orai2 (Abcam, ab180146), anti-Ampkα (Cell signaling, #2532), anti-phospho-AMPKα (cell signaling, #2535), anti-Stim1 (Abcam, ab108994), anti-Cul4A (Abcam, ab92554), anti-DDB1 (Bethyl Laboratories, A300-462A), anti-phospho-raptor (Cell signaling, #2083), anti-raptor (Cell signaling, #2280), anti-phospho-S6K (Cell signaling, #9206), anti-S6K (Cell signaling, #9202), anti-Lamin B (Santa Cruz, sc-374015), anti-Erk2 (Santa Cruz, sc-154), anti-CD16/32 (BioLegend, #101302), anti-CD11b (BioLegend, #1010207), and anti-β-Actin (Santa Cruz, sc-1616) were purchased.

**Mice**. $Crbn^{-/-}$ mice were a generous gift from Cheol-Seung Park at Gwangju Institute of Science and Technology (GIST) and described previously[27]. C57BL/C were purchased from Taconic bioscience. The mice were bred in equipped animal facility with temperature at 20–25 °C and humidity at 30–70%, under the same dark/light cycle (12:12). All experiments using mice were approved by the animal care and ethics committees of GIST in accordance with the national institutes of health guide for the care and use of laboratory animals.

**Immunoblotting and immunoprecipitation**. Primary or transfected cells were lysed using lysis buffer (50 mM Tris (pH 7.6), 150 mM NaCl, 10 mM NaPP, 10 mM NaF, 1 mM $Na_3VO_4$, 1% Triton X-100, 10 µg/mL pepstatin, 10 µg/mL leupeptin, 10 µg/mL AEBSF, and 10 µg/mL aprotinin). Proteins in the lysates were separated by SDS–PAGE, transferred onto a nitrocellulose membrane, and detected by appropriate antibodies. For immunoprecipitation, lysates were incubated with Flag M2 agarose beads (Sigma, A2220), Glutathione-Sepharose 4B beads (GE health-care, 17-0756-01), or protein A/G agarose beads (Santa Cruz, sc-2003) for 2 h. Proteins bound to the beads were separated by SDS–PAGE, transferred onto a nitrocellulose membrane, and detected by immunoblotting.

**Efferocytosis assay**. Efferocytosis assay was performed as described previously[53]. Briefly, LR73 cells, MEFs, BMDMs, or peritoneal macrophages were plated in a 24-well plate. One day after plating, the cells were incubated with targets (TAMRA-stained apoptotic thymocytes, PS beads, or bioparticles) for the indicated times. Apoptotic thymocytes were generated by incubation of the cells with 50 µM dexamethasone at 37 °C for 4 h. For the generation of PS beads, 6 µm neutravidin beads (Spherotech, NVP-60-5) were coated with biotin-PS. Cy3-conjugated and biotin-conjugated 21-mer dsDNA were used to label the PS beads. After incubation of phagocytes with the targets, phagocytes were extensively washed with ice-cold PBS, trypsinized, and analyzed by flow cytometry (BD FACS Canto II). Data acquired by flow cytometry were analyzed using FlowJo software. MEFs or BMDMs were treated with 10 µM YM-58483 (Sigma, Y4895), 25–200 µM 9-AC (Sigma, A89405), 5 mM EGTA, 50 µM GSK-5498A (MedChemExpress, HY-12521), or 5–50 µM Mdivi-1 (Sigma, M0199). For in vivo analysis of efferocytosis. $1 \times 10^7$ TAMRA-stained apoptotic cells in 300 µl PBS were injected intraperitoneally into 5-week-old WT or $Crbn^{-/-}$ mice. 15 min after injection, the mice were euthanized and peritoneal exudates stained with FITC-conjugated F4/80 antibody (Biolegend, 123108). TAMRA and FITC double-positive cells were considered as peritoneal macrophages engulfing apoptotic cells. To evaluate apoptotic cell clearance in the thymus, 5-week-old WT and $Crbn^{-/-}$ mice were peritoneally injected with 300 µl PBS containing 250 µg dexamethasone. 4 h after the injection, the thymi from the mice were extracted, and the number of thymocytes was counted using Sphero AccuCount particles (Spherotech, ACBP-50-10) and flow cytometry. Gating strategies to determine the percentage of phagocytes engulfing apoptotic cells are provided in Supplementary Fig. 19a–c. In addition, terminal deoxynucleotidyl transferase dUTP nick end labeling (TUNEL) assay was performed. The thymi were embedded in optimal cutting temperature compound and frozen in liquid nitrogen. Cryo-sectioned thymus on slides were stained using In Situ Cell Death detection kit (Roche, 12156792910) and DAPI (Sigma, Duo82040) according to the manufacturer's instructions. Images of sectioned thymus were obtained by Axio Imager D2 (Zeiss, Jena, German).

**Immunofluorescence microcopy**. BMDMs were plated on 18 mm Ø coverslip, stained with 0.5 µM Celltracker (Thermo, C2925), and incubated with TAMRA-stained apoptotic thymocytes or Cy3-labeled PS beads for 30 min at 4 °C. After incubation, the cells were washed with ice-cold PBS, fixed in 4% paraformaldehyde, and stained by Hoechst 33342 (Invitrogen, H1399). Images were acquired using Axio Imager D2 (Zeiss, Jena, German).

**Time-lapse confocal microscopy**. BMDMs plated on 13 mm Ø confocal dish were stained with 0.5 µM CellTracker (Thermo, C2925) and SiR-actin (Cytoskeleton, CY-SC001) for 30 min. The cells were then incubated with Cy3-labeled or non-labeled PS beads in the presence or absence of 10 µM YM-58483. Immediately after adding the beads, the cells were observed with Olympus FV1000 SPC (Olympus, Tokyo, Japan). Images were taken every 20 s for 30 min. For phalloidin staining, BMDMs stained with 0.5 µM CellTracker were incubated with PS beads for 15 min, fixed in 4% paraformaldehyde, and permeabilized by 0.1% Triton X-100. Then, the cells were stained with Alexa Fluor 594-conjugated phalloidin (Life technology, A12381) and Hoechst 33342 (Invitrogen, H1399).

**Quantitative RT-PCR**. Total RNA was extracted from WT and Crbn$^{-/-}$ cells using RNeasy Plus Mini kit (Qiagen, 74136) and cDNA was synthesized using SuperScript IV First-Strand Synthesis System (Invitrogen, 18091200) according to the manufacturer's instructions. The relative levels of the transcripts of Orai1, Orai2, and Orai3 were analyzed using StepOnePlus real-time PCR system (Applied Biosystem).

**Migration assay**. BMDMs were maintained in RPMI containing 0.5% FBS for 4 h. RPMI containing 0.5% FBS mixed with 100 nM ATP or ATPγS were added to the lower chamber of a 24 transwell plate and $1 \times 10^5$ BMDM in RPMI containing 0.5% FBS were placed in the upper chamber of a transwell plate (Corning, CLS3422). 6 h after incubation, migrated BMDMs were fixed with 100% methanol for 20 min, stained with 0.5% crystal violet for 20 min, and observed by ZEISS Axio Vert.A1 (Zeiss, Jena, German).

**Measurement of calcium flux**. To measure SOCE, phagocytes were plated in a 96-well plate and stained with Fura2-AM (Invitrogen, F1221) in staining buffer (10 mM HEPES, 150 mM NaCl, 5 mM KCl, 0.1% glucose, 1% FBS, 2.5 mM probenecid, 1 mM CaCl$_2$, 1 mM MgCl$_2$, and pH 7.4). The phagocytes were treated with 1 μM thapsigargin (Sigma, T9033) and 2 mM CaCl$_2$ at the indicated times. The intensity of fluorescence from the cells was measured using a microplate reader (FlexStation 3, Molecular Devices). To evaluate the effects of the inhibitors on SOCE, phagocytes were preincubated with 10 μM YM-58483 (Sigma, Y4895) and SOCE was measured. Peak F340/380 of SOCE is the maximal ratio between calcium bound Fura2(F340, Ex/Em 340/510 nm) and unbound Fura2(F380, Ex/Em 380/510 nm). Slope is $(F_{max}-F_{360s})/(t_a-t_{360s})$, where $t_a$ is the time value that fluorescence of Fura-2 is maximal. Area under curve (AUC) of SOCE was calculated as follows:

$$AUC = \frac{1}{2}\sum_{i=1}^{n-1}(t_{i+1}-t_i)(F_{i+1}+F_i-2B),$$

where $t_i$ is the ith time values, $F_i$ the ith fluorescence value, $n$ is the number of time values, and $B$ is the initial fluorescence values of Fura-2 at 30 s. To measure intracellular calcium levels during efferocytosis, phagocytes were stained with Fluo3-AM (Invitrogen, F1242) or Fura2-AM in staining buffer (10 mM HEPES, 150 mM NaCl, 5 mM KCl, 0.1% glucose, 1% FBS, 2.5 mM probenecid 1 mM CaCl$_2$, 1 mM MgCl$_2$, and pH 7.4) for 30 min. After that, apoptotic thymocytes in DMEM with or without calcium were added to the phagocytes and the intensity of fluorescence of the cells was measured with a microplate reader (FlexStation 3, Molecular Devices).

**Yeast two-hybrid screen**. Yeast two-hybrid assay was performed as previously described[54]. Briefly, yeast cells (HF7C) were transformed with the indicated plasmids by the LiAc-based method and plated on the non-selective plates without Trp and Leu. The yeast clones growing on the non-selective plate were dotted on the selective plate containing 5 mM 3-amino-1,2,4-triazole (3AT) without His, Trp, and Leu.

**Statistical analysis**. Data are shown as the mean ± standard error of mean (SEM). All experiments were independently performed at least two times and statistical significance of difference was analyzed by unpaired Student's two-tailed $t$ test, One-way ANOVA, or two-way ANOVA using the GraphPad Prism 7 software. Significance was accepted when $p$ values were < 0.05.

**Reporting summary**. Further information on research design is available in the Nature Research Reporting Summary linked to this article.

## Data availability

Data supporting the findings of this study are available within the article and its Supplementary Information files or from the corresponding author upon reasonable request. Source data are provided with this paper.

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

## Acknowledgements

This work was supported by the National Research Foundation of Korea funded by the Korea government (MSIP) (2019R1A2C1006480, 2019R1I1A1A01057419, and 2019R1A4A1028802) and by Aging Research Institute at GIST.

## Author contributions

Conceptualization, H.M. and D.P.; Methodology, C.M., G.K., D.K., K.K., S.-A.L., B.M., and S.-J.Y.; Formal analysis, H.M., C.M., G.G., and D.P.; Resources, S.-J.Y., G.L., S.K.C., C.S.L., and C.-S.P.; Writing, H.M. and D.P.; Funding acquisition, J.L. and D.P.

## Competing interests

The authors declare no competing interests.
