## [Peer Review File · Nature Communications]

Reviewers' comments:

Reviewer #1 (Remarks to the Author):

In this work the authors provide evidence that *orai1* is a native substrate for the CRL4-CRBN ubiquitin ligase, with biological consequences during efferocytosis.

Moon et al. show that cereblon overexpression in the Chinese Hamster Ovary fibroblast line LR73 that cereblon overexpression correlates with reduced phagocytic engulfment. They show the converse in immortalized mouse MEF cells, where the cereblon knockout increases efficiency of efferocytosis. The authors conclude that cereblon negatively regulates phagocytosis. However, the comparison is not well controlled, comparing independently derived cell lines and comparing between species. The authors have further a large number of observations on the phenomenological differences on phagocytosis in vivo in the cereblon knockout mouse.

The native functions of cereblon are an interesting open question in biology. This work includes a large body of phenomenological observations from cereblon knockout mice and various cell lines. The effects reported in the manuscript may be measurable and significant, but they do not appear to be absolutely critical to the biological processes described. Furthermore, there is an incomplete set of experiments to provide robust molecular support for the underlying hypothesis, and the molecular basis for variable recruitment of *orai1* by cereblon during efferocytosis is not established in this work.

The identification of additional cereblon native substrates associated with biological effects would be of great interest to the scientific community. This work leaves some key questions unanswered and does not completely prove the link or explain the role of cereblon and *orai1*. This reviewer would recommend additional work before for reconsideration for publication.

Major Issues:

- Line 120, Figure 1a, 1b. The initial cellular work is the foundation for the large body of work that follows, however, key controls are missing and therefore the fundamental conclusions are not well supported by data:

- o Why was the phagocytosis measured for cereblon overexpression and knockout in different cell lines from different species (Chinese hamster and mouse)? To show a robust phenomenon, the effect of knockout and overexpression would be measured in a single system or both systems in parallel.

- o In the cereblon knockout MEFs, could the authors rescue the knockout phenotype with overexpression of cereblon to demonstrate that it is indeed loss of cereblon that drives the phenotype and not an artifact of deriving an independent cell line?

- Line 243. The authors again compare between LR73 Chinese Hamster cells overexpressing cereblon and mouse cells in which cereblon is knocked out. This is not a properly controlled observation, would it be possible to examine the phenotype of overexpression and knockout in the same line?

- The half life of *orai1* protein in the presence and absence of cereblon should be shown in the within one cellular system (eg HEK293).

- Line 273, it is true that inverse correlation of cereblon and *orai1* may suggest regulation, but it is not accurate that this alone implies that *orai1* is a direct substrate for CRL4CRBN. Please consider text changes here.

- The molecular basis for the reduced interaction between cereblon and *orai1* during efferocytosis is not established. Is this due to a post-translational modification? Different subcellular compartmentalization?

- Figure 4h and i. The structure of cereblon in complex with DDB1 has been elucidated (eg Fischer et al. Nature 2014, Chamberlain et al. NSMB 2014). How do the truncations map to the known domains and do they all preserve known DDB1 interactions? It would be helpful to see that the cereblon truncation mutants tested still precipitate the functional components of the CRL4 E3 ubiquitin ligase. This would show that the proteins are soluble, folded and correctly localized.

- If possible, an in vitro reconstitution of the binding and ubiquitination would provide strong support

for the proposed recruitment of orai1.

Reviewer #2 (Remarks to the Author):

In this manuscript, Moon et al. describe a novel regulatory mechanism of efferocytosis, in which the binding and subsequent ubiquitin-mediated degradation of the calcium channel Orai1 by Crbn, a component of the CRL4 E3 ubiquitin ligase, limits calcium influx (SOCE) in macrophages. The attenuated Orai1-mediated SOCE is recovered in macrophages during the phagocytosis of apoptotic material (efferocytosis) due to a reduction in the interaction between Orai1 and Crbn. Crbn knockout macrophages have enhanced SOCE and efferocytosis, with quicker phagocytic cup formation and uptake of apoptotic material. Finally, the defect in efferocytosis in Crbn overexpressing cells is rescued by overexpression of Orai1. Overall the study is methodical, well written and offers new insights into both the regulation of SOCE and the physiological role of SOCE in macrophages and efferocytosis. There are however several technical issues that need to be addressed and the overall significance of the Crbn-Orai1 interaction for efferocytosis in vivo needs to be established.

Specific comments:

Western blots for Orai1 (in main figures) and Orai2 in Suppl. Figure 5 lack specificity controls. Orai antibodies are of notoriously poor quality and it is unclear that the antibodies used here in fact recognize Orai1 and Orai2. There also are some discrepancies regarding the size of Orai1 bands detected in Fig.1a and 1b, and Fig.6b. The authors need to establish specificity using, for instance, RNAi methods. Please show entire Western blots for Orai1 and Crbn. Furthermore, the reported increases in Orai1 expression in Crbn^{-/-} cells (Fig.4a,b, Fig.4f, Fig.5d, Fig.6b) are fairly small and require a quantification of repeat experiments.

The authors nicely narrowed down Crbn binding to the N terminus of Orai1. Does Crbn also ubiquitinate the N terminus and if yes which lysine residue? This is an important question to answer in my opinion, because the authors could overexpress a form of Orai1 with a point mutation of this K residue to show that Crbn indeed regulates efferocytosis by modulating Orai1 ubiquitination and expression levels.

A previous study showed that Ubiquilin 1 increased the ubiquitination of Orai1, resulting in degradation of Orai1 and reduced SOCE (PMID: 23307288). In this study, the proteasome inhibitor MG132 failed to block the degradation of Orai1 (unlike in this manuscript by Moon et al) whereas the lysosome inhibitor bafilomycin A prevented Orai1 degradation. Indeed, cell surface expression of several ion channels (e.g. hERG) was shown to be regulated by monoubiquitination and degradation through the multivesicular body / lysosomal pathway. Can the authors comment on this difference please.

The authors show that expression of Orai1 (Fig.6a) and Orai2 (Suppl Fig.5e) enhances efferocytosis, whereas expression of Orai3 does not. This is consistent with results by Gronski et al. 2009 (PMID: 19461656), who showed that RNAi of Orai1 and Orai2 (and to a lesser degree Orai3) reduces efferocytosis. In this context it is interesting to comment that deletion of Orai2 in BMDM increases SOCE (PMID: 26109647), whereas deletion of Orai1 reduces it. This raises the questions how well SOCE correlates with efferocytosis. The authors should directly test the effects of Orai1 and Orai2 deletion on efferocytosis (similar to the deletion of Crbn in Fig.1).

It is noteworthy that inhibition of SOCE has no effect on many macrophage functions including phagocytosis (PMID: 26109647; PMID: 29176619). How do the authors explain that Ca²⁺ influx by SOCE has an effect on efferocytosis but not on other phagocytosis mechanisms?

The effects of Crbn and Orai1 expression on efferocytosis in vitro (Fig.6a) are fairly small. This makes me wonder how relevant the Crbn-Orai1 interaction is for efferocytosis and clearance of apoptotic cells in vivo. I think the authors need to show the significance of their findings in vivo, for instance in an autoimmune disease model for SLE or RA. One would expect that Crbn^{-/-} mice are protected from disease, at least partially.

Minor comments:

- It is noteworthy that for all IP and pulldown experiments, the IP/pulldown was done against Orai1 followed by detection of Crbn by WB. There is no experiment to do the reverse. Have the authors conducted experiments to first IP Crbn and then detect Orai1 by WB?
- Figure 1e: percent numbers in FACS plot (top) are too small to read.
- Figure 1g: Y-axis legend says TUNEL intensity, whereas in the text authors refer to "fewer TUNEL positive cells" (line 157)
- Figure 1i needs a better explanation why ATP stimulation in a transwell assay measures BMDM migration and is a relevant assay to the question of thymocyte clearance.
- Figure 3c needs a legend to explain what the 4 traces represent. More importantly, how to the authors interpret the residual SOCE in YM58483-treated cells? Is it due to Orai3, which is not modulated by Crbn?

Reviewer #3 (Remarks to the Author):

This study reports that cereblon (Crbn), a substrate acceptor of E3 ubiquitin ligase, negatively regulates the clearance of apoptotic cells by macrophages, efferocytosis, by targeting the store-operated Ca²⁺ channel ORAI1 for ubiquitin-mediated proteosomal degradation. Crbn-deficient macrophages had increased store-operated Ca²⁺ entry (SOCE) and captured apoptotic cells more rapidly, promoting efferocytosis, while Crbn overexpression had the opposite effects. Biochemically, Orai1 interacted with Crbn and was ubiquitinated and degraded in a Crbn-dependent manner. The Crbn-Orai1 interaction was reduced during efferocytosis, increasing the amount of Orai1 and facilitating phagocytosis.

Comments:

Orai1 channels are critical components of the store-operated Ca²⁺ entry (SOCE) pathway that control the function of immune cells, and patients with loss of function mutations in Orai1 suffer from severe combined immunodeficiency and autoimmunity. The report that cereblon targets Orai1 for ubiquitination and degradation is novel and important, and the observation that this new regulatory mechanism facilitates efferocytosis by macrophages confirms the role of Orai1-mediated Ca²⁺ signals in phagocytosis.

The biochemical data are very solid and clearly show that Crbn interacts with Orai1 to regulate its ubiquitination. The effects of Crbn overexpression and knockout on the efficiency of efferocytosis and on SOCE are adequately documented. I have requests for additional experiments and clarification to complete the study.

1. The co-IP experiments with truncated mutants indicate that Crbn interacts with the N-terminal residues 1-98 of Orai1 (Fig 4g-i and S5). Orai1 exist in two forms, short and long, originating from alternative translation initiation at residue M64 <https://www.ncbi.nlm.nih.gov/pubmed/22641696>. The short isoform (Orai1b) has higher mobility in the membrane, assessed by FRAP, and reduced fast Ca²⁺-dependent inactivation <https://www.ncbi.nlm.nih.gov/pmc/articles/PMC4583604/>, reviewed in <https://www.ncbi.nlm.nih.gov/pubmed/29217255>. If binding involves the 1-98 N-terminal residues, it is likely that only the long form is targeted for ubiquitination. This should be tested, by mutating the

first (M1A) and second methionine (M64A) as in <https://www.ncbi.nlm.nih.gov/pubmed/22641696>.

2. The implications of the regulation of Orai1 stability by Crbn go beyond the facilitation of efferocytosis. It would therefore be important to document whether this mechanism contributes to the regulation of Orai1 stability in tissues that predominantly rely on the STIM/ORAI pathway for signaling. Mice with a deletion in Crbn had increased T cell activation, but this phenotype was attributed to epigenetic regulation of Kv1.3 expression <https://www.ncbi.nlm.nih.gov/pubmed/27439875>. Since the authors have access to this murine model, they should check the expression levels and membrane stability of Orai1 in T cells from Crbn-deficient mice.

3. The authors rule out a role for AMPK in mediating the effects of Crbn, based on the lack of effect of constitutively active or dominant negative AMPK mutants on efferocytosis by LR73 cells (Fig. S2). The lack of effect of the mutants contradicts the author's postulate that Orai1 activity regulates efferocytosis, because AMPK has been shown to affect Orai1 protein stability <https://www.ncbi.nlm.nih.gov/pubmed/22682960>, <https://www.ncbi.nlm.nih.gov/pubmed/24080823>. These mutants are thus expected to impact SOCE, regardless of whether AMPK is implicated in mediating the effects of Crbn. This contradiction needs to be addressed, and controls included to document that the mutants used have altered kinase activity in LR73 cells.

Other comments:

Figs 1-3: The Ca²⁺ signaling and phagocytic functions of bone-marrow derived cells from WT and knock-out animals are compared. Controls are missing to document that the nature and differentiation state of these phagocytic cells are identical. Please show FACS profiles with appropriate markers to clarify this.

Fig. 2b: Instead of a linescan, the fluorescence intensity should be integrated over the entire periphagosomal area to better document the actin staining.

Fig. 2d, f: The effects of Orai1 channel inhibitors on the kinetics of phagocytic cup closure should be documented.

Fig. 2g, h: Why was Fluo3 used for these Ca²⁺ recordings and not a ratiometric dye as in the subsequent figure?

Fig. 3: The pyrazole compound used as SOCE channel inhibitor, BTP2, aka YM-58483, is not specific. It activates TRPM4 channels <https://www.ncbi.nlm.nih.gov/pubmed/16407466>, inhibits TRPC3 and TRPC5 channels <https://www.ncbi.nlm.nih.gov/pubmed/15647288> and binds to the actin reorganizing protein drebrin <https://www.ncbi.nlm.nih.gov/pubmed/19948240>. A more specific inhibitor such as GSK-5498A should be used to confirm the pharmacological findings.

Figs 4-5: The study would benefit from the inclusion of quantitative evaluations of the western blots

Fig 6A. Crbn expression is said to reduce the engulfment on apoptotic cells, but no significance is indicated in the graph between these conditions.

Fig. 6b: The increased Orai1 abundance in macrophages after 30-60 minutes of phagocytosis might reflect the detection of channel protein coming from internalized thymocytes. T cells express high amounts of Orai1 channels, whose activity is required for their proliferation. This could be tested by using thymocytes expressing a tagged version of the Orai1 channel as phagocytic prey.

Fig. 6d: related to the point above, the decreased Crbn binding to Orai1-FLAG might reflect scavenging of Crbn by Orai1 from thymocytes.

The interaction between Orai1 and Crbn may be indirect and it would be interesting to see if this interaction is complexed with other proteins.

Minor Comments:

In line 88-89 – ‘mental retardation’ is no longer used and was changed to ‘intellectual disability’ in 2013 (ref. https://journals.lww.com/co-psychiatry/FullText/2013/05000/New_terminology_for_mental_retardation_in_DSM_5.6.aspx)

Line 166 ‘though’ should be ‘through’

We would like to thank the reviewer, who provided constructive reviews of the manuscript. Their specialized knowledge helped generate more reliable data, find more effective ways of expressing our conclusion, and thus improve the rigor and clarity of the manuscript. We are also grateful for the opportunity for revision.

Response to Referee #1

Reviewer #1 (Remarks to the Author):

In this work the authors provide evidence that orai1 is a native substrate for the CRL4-CRBN ubiquitin ligase, with biological consequences during efferocytosis.

Moon et al. show that cereblon overexpression in the Chinese Hamster Ovary fibroblast line LR73 that cereblon overexpression correlates with reduced phagocytic engulfment. They show the converse in immortalized mouse MEF cells, where the cereblon knockout increases efficiency of efferocytosis. The authors conclude that cereblon negatively regulates phagocytosis. However, the comparison is not well controlled, comparing independently derived cell lines and comparing between species. The authors have further a large number of observations on the phenomenological differences on phagocytosis in vivo in the cereblon knockout mouse.

The native functions of cereblon are an interesting open question in biology. This work includes a large body of phenomenological observations from cereblon knockout mice and various cell lines. The effects reported in the manuscript may be measurable and significant, but they do not appear to be absolutely critical to the biological processes described. Furthermore, there is an incomplete set of experiments to provide robust molecular support for the underlying hypothesis, and the molecular basis for variable recruitment of orai1 by cereblon during efferocytosis is not established in this work.

The identification of additional cereblon native substrates associated with biological effects would be of great interest to the scientific community. This work leaves some key questions unanswered and does not completely prove the link or explain the role of cereblon and orai1. This reviewer would recommend additional work before for reconsideration for publication.

Major Issues:

- Line 120, Figure 1a, 1b. The initial cellular work is the foundation for the large body of work that follows, however, key controls are missing and therefore the fundamental conclusions are not well supported by data:

o Why was the phagocytosis measured for cereblon overexpression and knockout in different cell lines from different species (Chinese hamster and mouse)? To show a robust phenomenon, the effect of knockout and overexpression would be measured in a single system or both systems in parallel.

Because apoptotic cells are phagocytosed by almost all types of cells including macrophages, epithelial cells, and fibroblasts, many different types of phagocytes have been used to evaluate the effects of a gene on efferocytosis *in vitro*. Although LR73 cells are non-professional phagocytes and Chinese hamster ovary cells, they are highly phagocytic for apoptotic cells, express many engulfment proteins, and are easily transfected. Due to these advantages, we and several groups in the field have routinely used them to initially evaluate the effects of overexpression of a gene of interest on efferocytosis *in vitro*. In addition, using LR73 cells and other types of cells, we better understand whether the effects of a gene are limited to a specific cell type or not. However, we appreciate the reviewer's point. To relieve the reviewer's concern, we overexpressed Crbn in MEFs and evaluated efferocytosis by the cells overexpressing Crbn. Like Crbn overexpression in LR73 cells, efferocytosis by MEFs overexpressing Crbn was less efficient than control cells as measured by the percentage of cells that engulfed apoptotic cells or the MFI of engulfing phagocytes. A lower percentage of MEFs overexpressing Crbn than control MEFs ingested apoptotic cells and the MFI of MEFs engulfing apoptotic cells was also lower in MEFs overexpressing Crbn than in control MEFs (Supplementary Fig. 1a, b in the revised manuscript), suggesting that the effects of Crbn overexpression on efferocytosis is not cell-type specific but robust and general. We hope that this additional experiment relieves the reviewer's concern. In addition, due to the GFP co-transfection issues, we redid the experiment in Fig. 1a. LR73 cells transfected with HA-Crbn were incubated with TAMRA-stained apoptotic thymocytes, stained with an anti-HA antibody, and analyzed by flow cytometry. HA- and TAMRA-positive cells were considered as Crbn overexpressing phagocytes engulfing apoptotic cells in the revised manuscript (Please, also refer to the response to the 6th comment of reviewer #2).

o In the cereblon knockout MEFs, could the authors rescue the knockout phenotype with

overexpression of cereblon to demonstrate that it is indeed loss of cereblon that drives the phenotype and not an artifact of deriving an independent cell line?

We regret not to perform the experiment in the original manuscript and thank the reviewer for the suggested experiment. As the reviewer indicated, we redid the experiment including Crbn overexpression in *Crbn*^{-/-} MEFs and tested whether Crbn expression in *Crbn*^{-/-} MEFs could rescue the phenotype of efferocytosis by *Crbn*^{-/-} MEFs. The enhanced efferocytosis by *Crbn*^{-/-} MEFs was nullified upon Crbn overexpression and efferocytosis by *Crbn*^{-/-} MEFs overexpressing Crbn was comparable with that by WT MEFs (Fig. 1b in the revised manuscript), which suggests that the phenotype of efferocytosis by *Crbn*^{-/-} MEFs is caused by loss of Crbn. Once again, we thank the reviewer for suggesting the experiment further clarifying the effect of Crbn on efferocytosis.

- Line 243. The authors again compare between LR73 Chinese Hamster cells overexpressing cereblon and mouse cells in which cereblon is knocked out. This is not a properly controlled observation, would it be possible to examine the phenotype of overexpression and knockout in the same line?

As we mentioned in the previous response, the hamster cell line is highly phagocytic, expresses most of key genes involved in efferocytosis, is easily transfected, and thus is widely used to evaluate the effects of a gene of interest on efferocytosis. Results derived from the cell line are also generally accepted in the field. However, we used the cell line without deep consideration as the reviewer indicated above. We redid the experiment with MEFs overexpressing Crbn to measure SOCE as the reviewer suggested. The overall SOCE phenotype of MEFs overexpressing Crbn was not different from that of LR73 cells overexpressing Crbn although there are differences in the values. SOCE was reduced by 10% in Crbn-overexpressing MEFs and the rate of calcium entry was also decreased in these cells (Supplementary Fig. 10b in the revised manuscript). We hope that the experiment relieves the reviewer's concern.

- The half life of orai1 protein in the presence and absence of cereblon should be shown in the within one cellular system (eg HEK293).

As the reviewer indicated, we used WT and *Crbn*^{-/-} 293T cells to measure the half-life of Orai1. Both WT and *Crbn*^{-/-} 293T cells were incubated with cycloheximide and the levels of Orai1 in the cells were measured. The half-life of Orai1 in WT 293T cells was about 12 h, which is about 2 folds shorter than

that in *Crbn*^{-/-} 293T cells (Supplementary Fig. 17a in the revised manuscript), again, suggesting that the level of Orai1 is modulated by Crbn. We thank the reviewer for requesting the experiment strengthening our findings.

- Line 273, it is true that inverse correlation of cereblon and orai1 may suggest regulation, but it is not accurate that this alone implies that orai1 is a direct substrate for CRL4CRBN. Please consider text changes here.

We changed the sentence in the revised manuscript as follows

‘which strongly implies that Crbn regulates the level of Orai1 and that Orai1 is a substrate of Crbn. To test whether Orai1 is a substrate of Crbn’

→ ‘which strongly implies that Crbn is involved in regulating the level of Orai1. To test whether Orai1 could be regulated as a Crbn substrate’

We hope that the revised sentence is not too assertive in describing the relationship between Orai1 and Crbn.

- The molecular basis for the reduced interaction between cereblon and orai1 during efferocytosis is not established. Is this due to a post-translational modification? Different subcellular compartmentalization?

We really appreciate pointing out a molecular mechanism how interaction between Crbn and Orai1 is reduced during efferocytosis. Indeed, we discussed a possible molecular mechanism by which the interaction is attenuated during efferocytosis in the original manuscript (line 358-369). Initially, we thought that the level of Crbn was altered during efferocytosis. However, this was not the case as described in the original manuscript (Fig. 6d in the revised manuscript). As we discussed in the discussion of the original manuscript and the reviewer indicated, a protein modification or/and an alteration of subcellular localization of Crbn could affect the interaction of Crbn with Orai1. Thus, we first tested whether the subcellular localization of Crbn was altered during efferocytosis. We found that the levels of Crbn in the cytoplasm or in the nucleus in LR73 cells incubating apoptotic cells was unaltered (Fig. 6g in the revised manuscript). In addition, confocal microscopy also showed that apoptotic cell stimulation did not change the subcellular localization of Crbn (Fig. 6f in the revised manuscript). Second, we investigated tyrosine-phosphorylation of Crbn during efferocytosis. The level of

phosphorylation of Crbn in LR73 cells incubating with apoptotic cells was similar to those in control cells (Supplementary Fig. 14 in the revised manuscript). These data suggest that neither altered subcellular localization nor modification of Crbn is a cause to induce attenuated interaction between Crbn and Orai1 during efferocytosis.

Once again, as indicated in the discussion of the original manuscript but data were not shown, we found that interaction between Orai1 and Stim1 was induced upon apoptotic cell stimulation (Fig. 6h in the revised manuscript), which strongly suggests that interaction between Crbn and Orai1 could compete with interaction between Orai1 and Stim1 during efferocytosis. Thus, we further tested whether the interaction of Crbn with Orai1 could be affected by expression of Stim1. When three proteins, Crbn, Orai1, and Stim1 were co-overexpressed, the interaction of Crbn with Orai1 was substantially weakened (Fig. 6i in the revised manuscript). These data imply that Crbn competes with Stim1, which may possess higher binding affinity to Orai1, for binding to Orai1, resulting in attenuated interaction of Crbn with Orai1 during efferocytosis.

Currently, we are investigating a mechanism by which the interaction between Orai1 and Stim1 is induced during efferocytosis. Preliminary data indicate that Mertk, an engulfment receptor for apoptotic cells, activates the PLC-IP₃-IP₃R axis, resulting in calcium release from the ER and thus inducing SOCE mediated by the Orai1-Stim1 complex upon apoptotic cell stimulation. Because these are beyond the scope of this study, as a following study for this work, we would like to report the mechanism, an upstream signaling pathway inducing SOCE during efferocytosis.

- Figure 4h and i. The structure of cereblon in complex with DDB1 has been elucidated (eg Fischer et al. Nature 2014, Chamberlain et al. NSMB 2014). How do the truncations map to the known domains and do they all preserve known DDB1 interactions? It would be helpful to see that the cereblon truncation mutants tested still precipitate the functional components of the CRL4 E3 ubiquitin ligase. This would show that the proteins are soluble, folded and correctly localized.

As the schematic diagram shown in supplementary Fig. 12a in the revised manuscript, Crbn^{AN} contains residues 264-445 and Crbn^{Lon} contains residues 84-320. Thus, both truncation mutants contain the DDB1 binding region which is the HBD of Crbn (the helical bundle domain of Crbn, residues 186-317 (Fischer et al. Nature 2014); the Lon-like domain of Crbn, residues 76-318 (Chamberlain et al. NSMB 2014)).

Therefore, it is possible that the truncation mutants could still form a complex with the functional components of the ligase as the reviewer indicated. We thus tested whether the truncation mutants could form a complex with the CRL4 E3 ligase. DDB1 and Cul4A were robustly co-precipitated with both

mutants (Supplementary Fig. 12b in the revised manuscript), suggesting that gross deletion in the truncation mutants does not cause the related issues which the reviewer indicated.

- If possible, an in vitro reconstitution of the binding and ubiquitination would provide strong support for the proposed recruitment of orai1.

We further confirmed the interaction between Crbn and Orai1 using a yeast two-hybrid assay. Crbn orthologues are found in vertebrates and plants but have not been found in yeast. In addition, yeast is eukaryotic cells in which Crbn is translated more appropriately than in bacteria. Thus, using yeast is advantageous to validate whether the interaction is direct. The fragments whose interaction was confirmed in mammalian cells were cloned into yeast expression vectors and performed a yeast two-hybrid assay. Yeast transformants expressing Orai1^{N-term} and Crbn grew on the selective plate whereas transformants expressing only either Orai1^{N-term} or Crbn failed to grow on it (Fig. 4j in the revised manuscript), suggesting that Crbn interacts Orai1^{N-term}. Furthermore, we tested interaction between Crbn and ORAI1 β , an alternative translated form of ORAI1 in human and lacking residue 1-63 of ORAI1. Interaction of Crbn with ORAI1 β was drastically diminished compared with that between Crbn and ORAI1 (Supplementary Fig. 16a, please also refer to the response to the 1st comment of referee #3), supporting that Crbn interacts with Orai1 through the N-terminus of Orai1.

In addition, we mutated putative ubiquitinated residues of Orai1 and tested whether the mutations impeded Orai1 ubiquitination. Mutation of residue 89 to alanine showed the strong inhibition of Orai1 ubiquitination (Fig. 5d in the revised manuscript, please also refer to the response to the 2nd comment of referee #2). Taken together, these data indicate that Orai1 indeed interacts with Crbn and is ubiquitinated. We hope that these additional data give the reviewer confidence in our conclusion.

Response to Referee #2

Reviewer #2 (Remarks to the Author):

In this manuscript, Moon et al. describe a novel regulatory mechanism of efferocytosis, in which the binding and subsequent ubiquitin-mediated degradation of the calcium channel Orai1 by Crbn, a component of the CRL4 E3 ubiquitin ligase, limits calcium influx (SOCE) in macrophages. The attenuated Orai1-mediated SOCE is recovered in macrophages during the phagocytosis of apoptotic material (efferocytosis) due to a reduction in the interaction between Orai1 and Crbn. Crbn knockout macrophages have enhanced SOCE and efferocytosis, with quicker phagocytic cup formation and uptake of apoptotic material. Finally, the defect in efferocytosis in Crbn overexpressing cells is rescued by overexpression of Orai1. Overall the study is methodical, well written and offers new insights into both the regulation of SOCE and the physiological role of SOCE in macrophages and efferocytosis. There are however several technical issues that need to be addressed and the overall significance of the Crbn-Orai1 interaction for efferocytosis in vivo needs to be established.

Specific comments:

Western blots for Orai1 (in main figures) and Orai2 in Suppl. Figure 5 lack specificity controls. Orai antibodies are of notoriously poor quality and it is unclear that the antibodies used here in fact recognize Orai1 and Orai2. There also are some discrepancies regarding the size of Orai1 bands detected in Fig.1a and 1b, and Fig.6b. The authors need to establish specificity using, for instance, RNAi methods. Please show entire Western blots for Orai1 and Crbn. Furthermore, the reported increases in Orai1 expression in Crbn^{-/-} cells (Fig.4a,b, Fig.4f, Fig.5d, Fig.6b) are fairly small and require a quantification of repeat experiments.

We agree that commercially available antibodies against Orai1 and Orai2 are of very poor quality. Even at overexpression conditions, most of the antibodies did not generate a clear Orai1 band. Detection of overexpressed Orai1-FLAG or precipitated Orai1-FLAG using an anti-FLAG antibody was unclear and generated even smear and messy bands as well. Besides the poor quality of Orai1 antibodies, the smear and messy band pattern of Orai1 likely results from properties of membrane proteins. Many membrane proteins indeed exhibit the band pattern similar to that of Orai1. After several antibody trials and optimization for western blotting (e.g. 37 °C incubation instead of boiling), we could find a couple of antibodies, which could detect endogenous Orai1 relatively well, from Santa Cruz (SC-68895) and

Alomone (ACC062). Especially, the antibody from Alomone was superior to detect Orai1 in BMDM and T cells.

As the reviewer suggested, we further tested the specificity of Orai1 antibody by depleting *Orai1* using siRNA. Based on RT-PCR, we could knockdown *Orai1* by 55% of using siRNA. Orai1 in MEFs transfected with siRNA decreased by 50% compared to that in control MEFs, which was commensurate with the levels of *Orai1* transcripts in MEFs transfected with *Orai1* siRNA (Supplementary Fig. 15e in the revised manuscript). In addition, we also compared the size of overexpressed Orai1 with that of endogenous Orai1. Both overexpressed and endogenous Orai1 were detected at the same location. The subtle difference in size could result from the FLAG tag and more amino acids in mouse Orai1 (mouse Orai1-304 amino acids vs human ORAI1-301 amino acids) (Data shown below). In a same way, Orai2 in MEFs transfected with *Orai2* siRNA was lower than that in control MEFs, which also corresponded to the decreased *Orai2* transcripts in MEFs transfected *Orai2* siRNA (Supplementary Fig. 15e in the revised manuscript). We think that these would alleviate the reviewer's concern about the quality of the antibodies.

The size marker for the Orai1 blot in Fig 6b was mislabeled. We confused the size marker of Orai1 with that of Crbn whose size is about 55 kDa. We are sorry for not being careful. We corrected the size marker in the revised manuscript.

As the reviewer suggested, the levels of Orai1 in the indicated figures were quantified (Fig. 5e and Supplementary Fig. 11a-c, 13a in the revised manuscript). Quantification apparently shows notable increase of Orai1 in Crbn deficient cells. All uncropped blots are shown in Data Source.

The authors nicely narrowed down Crbn binding to the N terminus of Orai1. Does Crbn also ubiquitinate the N terminus and if yes which lysine residue? This is an important question to answer in my opinion, because the authors could overexpress a form of Orai1 with a point mutation of this K residue to show that Crbn indeed regulates efferocytosis by modulating Orai1 ubiquitination and expression levels.

As the reviewer suggested, we mutated the lysine residues in the N-terminus of Orai1 and also the other lysine residues which located in the C-terminus and the intracellular loop of Orai1. As the reviewer anticipated, mutation of residue 89 to alanine showed the strongest inhibition of Orai1 ubiquitination and mutation of residue 163 or 268 marginally reduced Orai1 ubiquitination but not others (Fig. 5d in the revised manuscript). These data, as the reviewer indicated, strengthen the notion that Orai1 is indeed ubiquitinated, which is modulated by Crbn during efferocytosis.

A previous study showed that Ubiquilin 1 increased the ubiquitination of Orai1, resulting in degradation of Orai1 and reduced SOCE (PMID: 23307288). In this study, the proteasome inhibitor MG132 failed to block the degradation of Orai1 (unlike in this manuscript by Moon et al) whereas the lysosome inhibitor bafilomycin A prevented Orai1 degradation. Indeed, cell surface expression of several ion channels (e.g. hERG) was shown to be regulated by monoubiquitination and degradation through the multivesicular body / lysosomal pathway. Can the authors comment on this difference please.

We also found the paper when we initially studied Orai1 as a Crbn interacting protein. However, we could not reproduce that bafilomycin A inhibits Orai1 degradation. In contrast, we repeatedly found that the level of Orai1 increased in the presence of MG132. The disparity of the results may result from different experimental conditions. For example, HEK 293 cells were treated with MG 132 for 4 h in the paper. However, we treated BMDMs with MG132 for 6 h (Fig. 5e in the revised manuscript). Thus, different cell types, species, and incubation times may result in the discrepancy. However, although we used the same experimental conditions, e.g. 9 h incubation and 293T cells, we could not reproduce the effect of bafilomycin A on Orai1. Slightly lower expression of Orai1 and beta-actin in cells treated with bafilomycin than in control cells might be due to the toxic effect of the drug (Data shown below). In addition, the figures and the legends in the paper are unclear and coarse, which makes the data unreliable. Furthermore, Orai1 seems to be polyubiquitinated rather than mono-ubiquitinated because Orai1 ubiquitination blots always show smeared Orai1 bands, indicating polyubiquitination of Orai1.

Additionally, as we discussed in the original manuscript, it is already known that ClC-1/2, a chloride ion channel and a substrate of Crbn, is ubiquitinated and degraded through the ubiquitin-proteasome pathway. In addition, Slo, a potassium channel and a substrate of Crbn, is polyubiquitinated and its cell surface expression is regulated by the ubiquitination. Various membrane proteins and channel proteins are also ubiquitinated and degraded through the ubiquitin-proteasome pathway, and especially E3 ligases such as Nedd4-2 and c-Cbl are involved in channel and membrane protein ubiquitination (PMID: 10074483; PMID: 15123669; PMID: 23348737; PMID: 19710010; PMID: 20525683; PMID: 24553136). Therefore, it seems that some ion channels are regulated by the pathway indicated by the reviewer while others could be modulated by the ubiquitin-proteasome pathway. As we indicated in the discussion part of the original manuscript, how those proteins are differently regulated by mono- and polyubiquitination remains elusive and needs to be addressed in future studies.

The authors show that expression of Orai1 (Fig.6a) and Orai2 (Suppl Fig.5e) enhances efferocytosis, whereas expression of Orai3 does not. This is consistent with results by Gronski et al. 2009 (PMID: 19461656), who showed that RNAi of Orai1 and Orai2 (and to a lesser degree Orai3) reduces efferocytosis. In this context it is interesting to comment that deletion of Orai2 in BMDM increases SOCE (PMID: 26109647), whereas deletion of Orai1 reduces it. This raises the questions how well SOCE correlates with efferocytosis. The authors should directly test the effects of Orai1 and Orai2 deletion on efferocytosis (similar to the deletion of Crbn in Fig.1).

We really appreciate the question raised by the reviewer. First of all, there are some differences between ours and the 2009 study by Gronski et al. In the 2009 study, the authors knockdowned all three OraIs and evaluated the effects of Orai knockdown on efferocytosis. Knockdown of all three homologues decreased efferocytosis with a difference in the degree of reduction; *Orai3* knockdown decreased efferocytosis by about 30%. However, in our study, we instead overexpressed Orai homologues and evaluated the effects of their overexpression on efferocytosis. *Orai3* overexpression

did not affect efferocytosis, which is also different from the 2009 study. In addition, the increased efferocytosis by LR73 cells overexpressing *Orai1* or *Orai2* was not as notable as the decreased efferocytosis by NIH3T3 cells transfected with *Orai1* or *Orai2* siRNA. Indeed, strictly speaking, the results are not consistent with ours.

Overexpression of a gene generally causes more artifacts than knockdown of the gene, which means that a knockdown experiment is more reliable than an overexpression experiment to evaluate the roles of a gene. Nevertheless, we did not perform the knockdown experiments. Therefore, as the reviewer suggested, we re-evaluated the effects of *Orai1* and *Orai2* on efferocytosis by depleting them using siRNA. Efferocytosis by MEFs transfected with *Orai1* siRNA decreased compared with that by control cells. Intriguingly, however, *Orai2* knockdown in MEFs unaltered efferocytosis (Supplementary Fig. 15e, f in the revised manuscript). As the reviewer indicated, it is reportedly that *Orai1* knockout decreases SOCE, but *Orai2* knockout increases SOCE instead of reducing it. Indeed, we observed decreased SOCE in MEFs transfected with *Orai1* siRNA, but not in MEFs transfected with *Orai2* siRNA (Supplementary Fig. 15g in the revised manuscript), which can explain the efferocytosis by MEFs transfected with *Orai1* or *Orai2* siRNA. However, due to the incomplete knockdown of *Orai2*, *Orai2* knockdown by siRNA may not correctly reflect the effects of *Orai2* depletion on SOCE and efferocytosis. Nonetheless, these data support that SOCE affects and correlates with efferocytosis.

The disparity between ours and the 2009 study may result from extracellular calcium concentrations. It is reported that SOCE is reduced in *Orai2* KO cells at a low extracellular calcium concentration (PMID: 29604961). It is possible that the authors in the 2009 study performed the experiment at a low calcium concentration, which might result in decreased efferocytosis. However, this may not be the case because a normal medium contains a high concentration of calcium (normally 1.8 mM), and a specific condition for the experiment is not described in the paper.

It is noteworthy that inhibition of SOCE has no effect on many macrophage functions including phagocytosis (PMID: 26109647; PMID: 29176619). How do the authors explain that Ca²⁺ influx by SOCE has an effect on efferocytosis but not on other phagocytosis mechanisms?

Efferocytosis is a type of phagocytosis, and efferocytosis is similar to phagocytosis in terms of phagocytosing particles. However, efferocytosis is quite different from phagocytosis in various aspects. Ligands, receptors, types of targets, target sizes and signaling molecules involved in efferocytosis differ from those in phagocytosis. In addition, post engulfment responses are also different; by producing

different types of cytokines, efferocytosis provokes anti-inflammatory responses, but phagocytosis elicits pro-inflammatory responses. It is reasonable that different signals generate different responses. We think that different signal transduction through different signal molecules in efferocytosis from that in phagocytosis determines whether SOCE is required. Specifically, apoptotic cells are recognized and engulfed through interaction ligands (phosphatidylserine) on apoptotic cells and receptors (phosphatidylserine receptors) on phagocytes in efferocytosis. This initial ligand-receptor signaling in efferocytosis is completely different from that in phagocytosis, which may result in necessity of SOCE. In addition, apoptotic cells are large particles compared with particles ingested through phagocytosis, and phagocytes need to engulf apoptotic cells as large as themselves in some cases, which might require and activate SOCE.

We also tested whether Crbn could affect phagocytosis of polystyrene beads and bioparticles such as *E. coli* and *S. aureus* particles. Phagocytosis of those particles by *Crbn*^{-/-} MEFs was comparable with that by WT MEFs (Supplementary Fig. 4a, b in the revised manuscript), suggest that Crbn-mediated calcium flux is not involved in phagocytosis of those particles. In addition, we found that interaction between Orai1 and Stim1 was induced in phagocytes stimulated with apoptotic cells, and that Mertk, an engulfment receptor for apoptotic cells, is involved in induction of the interaction through the PLC-IP₃-IP₃R pathway releasing calcium from ER and thus causing interaction of Orai1 with Stim1 (Fig. 6h in the revised manuscript and unpublished data, please also refer to the response to the 6th comment of reviewer #1). It seems that signaling in efferocytosis cross-talks with signaling for SOCE but not signaling in phagocytosis of non-apoptotic cell targets. In sum, efferocytosis but not phagocytosis could activate signaling for SOCE and requires calcium influx through the Orai1-Stim1 complex. We hope that these are the answers to the question.

The effects of Crnb and Orai1 expression on efferocytosis in vitro (Fig.6a) are fairly small. This makes me wonder how relevant the Crbn-Orai1 interaction is for efferocytosis and clearance of apoptotic cells in vivo. I think the authors need to show the significance of their findings in vivo, for instance in an autoimmune disease model for SLE or RA. One would expect that Crbn^{-/-} mice are protected from disease, at least partially.

The small effect of Crbn or Crbn and Orai1 on efferocytosis in Fig 6a might be caused by the way for an efferocytosis assay. To evaluate the effect of a gene on efferocytosis, in some case we co-transfect phagocytes with a gene and GFP, incubate the phagocytes with TAMRA-stained apoptotic cells, and analyze the cells using flow cytometry. We consider GFP- and TAMRA-positive cells as phagocytes

engulfing apoptotic cells. Thus, it is considered that GFP-positive phagocytes as phagocytes expressing the gene. Sometimes, although this experimental way is useful, it underestimates the effect of the gene on efferocytosis because GFP-positive phagocytes not always express the gene. Especially, when transfection efficiency is low, the possibility that GFP-positive cells express the gene is low. The effect of Crbn overexpression on efferocytosis in Fig. 6a was evaluated in this way. Thus, the small effect of Crbn on efferocytosis in Fig. 6a resulted from gating GFP-positive cells rather than Crbn-positive cells and low transfection efficiency (Fig. 1a in the original manuscript shows the more apparent effect of Crbn than Fig. 6a due to high transfection efficiency).

Thus, we re-evaluate the effects of Crbn on efferocytosis by gating Crbn-positive cells. LR73 cells transfected with HA-Crbn were incubated with TAMRA-stained apoptotic cells, incubated with an FITC conjugated anti-HA antibody and analyzed using flow cytometry. HA- and TAMRA-positive cells were considered as Crbn overexpressing phagocytes engulfing apoptotic cells. The effect of Crbn overexpression on efferocytosis was prominent, and Orai1 co-expression with Crbn conspicuously rescued the phenotype of efferocytosis by phagocytes overexpressing Crbn (Fig. 6a in the revised manuscript).

Although the p value was calculated between control (GFP in the original manuscript) and Crbn and it is statistically significant in the original manuscript, we did not put the p value in the original Fig. 6a because it was shown in Fig. 1a. However, we were not careful and we should have put the value. In the revised version, we redid the experiment in Fig. 6a and put the p value between control and Crbn. We hope that the new data relieve the reviewer's concern.

We evaluated the effect of Crbn on efferocytosis using two different approaches *in vivo*, clearance of apoptotic cells in the thymus and peritoneum and the data indicate that Crbn modulated clearance of apoptotic cells *in vivo* as well, which is phenomenal and commensurate with *in vitro* data. The current scope of our study is to delineate a molecular mechanism by which Crbn modulates clearance of apoptotic cells. We think that validating the roles of Crbn at a pathological condition could be performed as a separate study in the near future and is beyond the current scope of this study. We hope that the provided *in vitro* and *in vivo* data could convince the reviewer of our conclusion.

Minor comments:

- *It is noteworthy that for all IP and pulldown experiments, the IP/pulldown was done against Orai1 followed by detection of Crbn by WB. There is no experiment to do the*

reverse. Have the authors conducted experiments to first IP Crbn and then detect Orai1 by WB?

We also performed the IP experiments in the inverse way. At an overexpression condition, Crbn-FLAG immunoprecipitation with anti-FLAG antibody-conjugated agarose beads could co-precipitate GST-Orai1, but co-precipitated GST-Orai1 with Crbn was not as clear and strong as the inverse way; a certain level of Orai1 was precipitated in a negative control due to stickiness of Orai1 (Data shown below). Thus, we decided to show co-precipitated Crbn with Orai1. In case of the IP experiment at endogenous protein levels, we failed to observe co-precipitated endogenous Orai1 with endogenous Crbn using an anti-Crbn antibody. Sometimes, an antibody could interfere with protein-protein interaction due to steric hinderance caused by it. Thus, protein-protein interaction is not always detected in both directions. We think that this is the case for the interaction of Crbn with Orai1. Because the anti-Crbn antibody which could detect and precipitate endogenous Crbn is only one which we have, we could not test it further. However, there was apparent interaction between Crbn and Orai1 at an endogenous level in the reverse way (Fig. 4f in the revised manuscript). In addition, the interaction was further confirmed using the yeast two-hybrid method, and interaction between Crbn and ORAI1 β , an alternative translated form of ORAI1 and lacking residue 1-63 of ORAI1, was noticeably reduced (Fig. 4j and Supplementary Fig. 16a, please refer to the response to the 1st comment of reviewer #3). Thus, we are sure that there is no doubt about the interaction between Orai1 and Crbn.

• *Figure 1e: percent numbers in FACS plot (top) are too small to read.*

As the reviewer requested, we increased the numbers in the dot plots, which improves readability.

- *Figure 1g: Y-axis legend says TUNEL intensity, whereas in the text authors refer to "fewer TUNEL positive cells" (line 157)*

We thank the reviewer for pointing it out. In the revised manuscript, we changed the sentence as follows. 'the intensity of TUNEL (TdT-mediated dUTP nick end labeling) was lower in *Crbn*^{-/-} mice than in *WT* mice following dexamethasone injection'

- *Figure 1i needs a better explanation why ATP stimulation in a transwell assay measures BMDM migration and is a relevant assay to the question of thymocyte clearance.*

We are sorry for not fully explaining the purpose of the migration experiment. The altered number of apoptotic cells *in vivo* could be caused by mainly three factors, the defect of efferocytosis, the different rate of apoptosis, and/or the different rate of phagocyte recruitment. A different rate of phagocyte migration affects the number of phagocytes at a site where apoptotic cells are generated or exist, which eventually affects removal of apoptotic cells. In order to show that the smaller number of apoptotic cells is caused by the promoted efferocytosis in *Crbn*^{-/-} mice, it should be shown that the rate of apoptosis and the rate of phagocyte recruitment are comparable between *WT* and *Crbn*^{-/-} mice. Apoptotic cells secrete chemoattractants called 'find-me signals', which include nucleotides (ATP and UTP), LPC, and S1P, to actively recruit phagocytes to them. ATP is a best-known chemoattractant secreted from apoptotic cells. That is why we performed the assay for the migration of BMDMs to ATP. To evaluate efferocytosis *in vivo*, these two control experiments are routinely performed in the field.

- *Figure 3c needs a legend to explain what the 4 traces represent. More importantly, how to the authors interpret the residual SOCE in YM58483-treated cells? Is it due to Orai3, which is not modulated by Crbn?*

We observed interaction between *Crbn* and *Orai3* at an overexpression condition (Supplementary Fig. 15c). Thus, it is possible that *Orai3* might be regulated by *Crbn* like *Orai1* and, if it is regulated by *Crbn*, the residual SOCE might not be due to *Orai3*. In addition, because YM-58483 similarly and substantially inhibited SOCE in both *WT* and *Crbn*^{-/-} BMDMs, and SOCE was comparable in *WT* and *Crbn*^{-/-} BMDMs in

the presence of YM-58483, the residual SOCE unlikely resulted from a molecule not regulated by Crbn. Instead, it seems that the residual SOCE in cells treated with YM-58483 in Fig. 3c likely resulted from partial inhibition because SOCE was reduced by YM-58483 in a dose-dependent manner (data shown below). We used four different colors to clearly distinguish the 4 traces in the revised manuscript, which improves readability. We hope that the reviewer will be satisfy with the changes.

Response to Referee #3

Reviewer #3 (Remarks to the Author):

This study reports that cereblon (Crbn), a substrate acceptor of E3 ubiquitin ligase, negatively regulates the clearance of apoptotic cells by macrophages, efferocytosis, by targeting the store-operated Ca²⁺ channel ORAI1 for ubiquitin-mediated proteosomal degradation. Crbn-deficient macrophages had increased store-operated Ca²⁺ entry (SOCE) and captured apoptotic cells more rapidly, promoting efferocytosis, while Crbn overexpression had the opposite effects. Biochemically, Orai1 interacted with Crbn and was ubiquitinated and degraded in a Crbn-dependent manner. The Crbn-Orai1 interaction was reduced during efferocytosis, increasing the amount of Orai1 and facilitating phagocytosis.

Comments:

Orai1 channels are critical components of the store-operated Ca²⁺ entry (SOCE) pathway that control the function of immune cells, and patients with loss of function mutations in Orai1 suffer from severe combined immunodeficiency and autoimmunity. The report that cereblon targets Orai1 for ubiquitination and degradation is novel and important, and the observation that this new regulatory mechanism facilitate efferocytosis by macrophages confirms the role of Orai1-mediated Ca²⁺ signals in phagocytosis.

The biochemical data are very solid and clearly show that Crbn interacts with Orai1 to regulate its ubiquitination. The effects of Crbn overexpression and knockout on the efficiency of efferocytosis and on SOCE are adequately documented. I have requests for additional experiments and clarification to complete the study.

1. The co-IP experiments with truncated mutants indicate that Crbn interacts with the N-terminal residues 1-98 of Orai1 (Fig 4g-i and S5). Orai1 exist in two forms, short and long, originating from alternative translation initiation at residue M64 <https://www.ncbi.nlm.nih.gov/pubmed/22641696>. The short isoform (Orai1b) has higher mobility in the membrane, assessed by FRAP, and reduced fast Ca²⁺-dependent inactivation <https://www.ncbi.nlm.nih.gov/pmc/articles/PMC4583604/>, reviewed in <https://www.ncbi.nlm.nih.gov/pubmed/29217255>. If binding involves the 1-98 N-terminal residues, it is likely that only the long form is targeted for ubiquitination. This should be tested, by mutating the first (M1A) and second methionine (M64A) as in <https://www.ncbi.nlm.nih.gov/pubmed/22641696>.

We thank the reviewer for pointing out the isoform. First of all, Orai1 and its homologs used in the study are mouse Orais. Although the data were not shown in the original manuscript, we had also confirmed interaction between Crbn and human ORAI1. To address whether the short isoform of ORAI1 (ORAI1 β), we directly introduced residues 64-301 (the short isoform) of human ORAI1 into a plasmid instead of using M1A and M64A and tested interaction between Crbn and ORAI1 β . Interaction of Crbn with ORAI1 β was drastically reduced compared with that of Crbn with ORAI1 although the interaction was not completely abolished (Supplementary Fig. 16a in the revised manuscript). Again, these data support that Crbn interacts with Orai1 through the N-terminus of Orai1.

Next, we measured ORAI1 β ubiquitination. Unexpectedly, ORAI1 β was still ubiquitinated and the ubiquitination level of ORAI1 β was comparable with that of ORAI1 (Supplementary Fig. 16b in the revised manuscript). At this point, it is unclear why ORAI1 β is still ubiquitinated although interaction between Crbn and ORAI1 β was almost abrogated. It is possible that the residual interaction of Crbn with ORAI1 β may be enough to ubiquitinate ORAI1 β . Another possibility is that ORAI1 β is ubiquitinated by a different E3 ligase but not the CRL4^{CRBN} E3 ligase. The latter is more plausible than the former because ORAI1 β was still ubiquitinated in *Crbn*^{-/-} 293T cells (Supplementary Fig. 16c). As the reviewer indicated and data recently published in Nature communications (PMID:31036819), CDI (calcium-dependent inactivation) of ORAI1 β is slower than that of ORAI1, which results in different calcium signals and NFAT activation by ORAI1 from those by ORAI1 β . Therefore, a different regulatory mechanism/a different E3 ligase by which the level of ORAI1 β is regulated may exist. We discussed these and cited papers related to the short form of ORAI1 β in the revised manuscript.

2. The implications of the regulation of Orai1 stability by Crbn go beyond the facilitation of efferocytosis. It would therefore be important to document whether this mechanism contributes to the regulation of Orai1 stability in tissues that predominantly rely on the STIM/ORAI pathway for signaling. Mice with a deletion in Crbn had increased T cell activation, but this phenotype was attributed to epigenetic regulation of Kv1.3 expression
<https://www.ncbi.nlm.nih.gov/pubmed/27439875>. Since the authors have access to this murine model, they should check the expression levels and membrane stability of Orai1 in T cells from Crbn-deficient mice.

We measured the level of Orai1 in CD4⁺ T cells isolated from the spleen of *WT* or *Crbn*^{-/-} mice. The level of Orai1 in CD4⁺ T derived from *Crbn*^{-/-} mice was comparable with that in CD4⁺ T derived from *WT* mice (Data shown below, left). In addition, when the lysates of CD4⁺ T cells were fractionized into two parts, the membrane fraction and cytoplasmic fraction, Orai1 was undetectable in both fractions. We tried to detect Orai1 in the fractions several times but eventually failed to detect it. This might be due to the very small cytoplasm of CD4⁺ T cells and the loss of proteins during preparation (Data shown below, right). Based on the level of Orai1 in the total cell lysate, the membrane stability of Orai1 in CD4⁺ T cells is unlikely regulated by *Crbn* at the basal state and that *Crbn* may regulate the level of Orai1 in a context-dependent manner. Indeed, the authors in the paper (PMID: 27439875) found that *Crbn* interacted with EZH1, a histone methyltransferase, and formed a complex with DDB1, Cul4A and EZH1 but the level of EZH1 in cells derived from *Crbn*^{-/-} mice was similar to that of EZH1 in cells derived from *WT* mice, also supporting context-dependent regulation by the CRL4^{CRBN} E3 ubiquitin ligase. Thus, there is a possibility that *Crbn*-Orai1 interaction in CD4⁺ T cells is induced by a certain stimulus and thus the level of Orai1 in CD4⁺ T cells may be altered by *Crbn* at a specific condition.

3. The authors rule out a role for AMPK in mediating the effects of Crbn, based on the lack of effect of constitutively active or dominant negative AMPK mutants on efferocytosis by LR73 cells (Fig. S2). The lack of effect of the mutants contradicts the author's postulate that Orai1 activity regulates efferocytosis, because AMPK has been shown to affect Orai1 protein stability
<https://www.ncbi.nlm.nih.gov/pubmed/22682960>, <https://www.ncbi.nlm.nih.gov/pubmed/24080823>.
These mutants are thus expected to impact SOCE, regardless of whether AMPK is implicated in mediating the effects of Crbn. This contradiction needs to be addressed, and controls included to document that the mutants used have altered kinase activity in LR73 cells.

The papers which the reviewer indicated show that *Ampk* negatively regulates the level of Orai1. Thus, it is presumed that the levels of Orai1 and SOCE in *Ampk*^{-/-} cells are higher than those in *WT* cells. We and previous studies clearly showed that the activation of *Ampk* in *Crbn*^{-/-} cells are higher than those in *WT*

cells. According to the logic of the papers, the levels of Orai1 and SOCE in *Crbn*^{-/-} cells should be lower than those in *WT* cells. However, we observed that the levels of Orai1 and SOCE in *Crbn*^{-/-} cells apparently increased. Indeed, the latter paper (PMID: 24080823) shows that Ampk deletion increased the level of Orai1 and SOCE in T cells. However, whether Ampk activation (a constitutive active form) or inactivation (a dominant negative form) alters the level of Orai1 had never been tested. Therefore, it is uncertain whether the level of Orai1 inversely corresponds to the status of Ampk. In other words, Ampk deletion may not inversely correspond to the activation of Ampk.

As the reviewer suggested, we tested whether the expression of a constitutive active (CA) form of Ampk indeed activates downstream signaling, which results in alteration of Orai1 levels. LR73 cells were transiently transfected with Ampk α CA and the phosphorylation levels of **S6k and Raptor** as well as the levels of Orai1 were measured. Raptor phosphorylation increased whereas S6k phosphorylation decreased, which suggests that Ampk α CA indeed activates the Ampk signaling pathway. In contrast, the levels of Orai1 was not changed at the condition (**Supplementary Fig. 5b in the revised manuscript**). Thus, Ampk activation does not change the level of Orai1, and increased SOCE in *Crbn*^{-/-} phagocytes is not related to Ampk activation. We hope that these solve the question that the reviewer raised.

Other comments:

Figs 1-3: The Ca²⁺ signaling and phagocytic functions of bone-marrow derived cells from WT and knock-out animals are compared. Controls are missing to document that the nature and differentiation state of these phagocytic cells are identical. Please show FACS profiles with appropriate markers to clarify this.

We thank the reviewer for pointing out the critical control. Because we differentiated bone marrow cells into macrophages, we used two makers, CD11b and F4/80, to evaluate the degree of differentiation. Double positive cells for CD11b and F4/80 in BMDMs derived from *Crbn*^{-/-} mice were comparable with those in BMDMs derived from *WT* mice (**Supplementary Fig. 3a, b in the revised manuscript**), suggesting that the enhanced efferocytosis by BMDMs derived from *Crbn*^{-/-} is not due to a difference in the degree of differentiation between *Crbn*^{-/-} and *WT* BMDMs.

Fig. 2b: Instead of a linescan, the fluorescence intensity should be integrated over the entire periphagosomal area to better document the actin staining.

At the reviewer suggested, we measured the fluorescence intensity of F-actin around the targets being engulfed (**Supplementary Fig. 7 in the revised manuscript**). Similar to the linescan data in the original manuscript, there was no difference of the F-actin intensity between BMDMs derived from *Crbn*^{-/-} and *WT* mice.

Fig. 2d, f: The effects of Orai1 channel inhibitors on the kinetics of phagocytic cup closure should be documented.

We thank the reviewer for suggesting a crucial control experiment, which indeed strengthens our conclusions. As the experiments in Figure 2d and 2f in the original manuscript, the phagocytic cup closure was monitored in the presence of YM-58483. The time for required for phagocytic cup closure during efferocytosis by *Crbn*^{-/-} and *WT* BMDMs were delayed in the presence of the inhibitor. In addition, the time for required for phagocytic cup closure during efferocytosis by *Crbn*^{-/-} BMDMs was increased more and thus indistinguishable from that during efferocytosis by *WT* BMDMs when the phagocytes were treated with the inhibitor (**Fig. 3f, g and Supplementary movie 4, 5 in the revised manuscript**). Once again, the data support that CRAC channels mediate the effects of *Crbn* on efferocytosis.

Fig. 2g, h: Why was Fluo3 used for these Ca²⁺ recordings and not a ratiometric dye as in the subsequent figure?

At the beginning of the study, we referred to several papers in that Fluo dyes were used to measure intracellular calcium levels during efferocytosis. Later, we realized that both Fluo3 and Fura2 are effective at detecting calcium, but Fura2 produced clearer data for SOCE than Fluo3. Thus, we used Fura2 for the experiments detecting SOCE. However, Fura2 was interchangeable with Fluo3 for monitoring intracellular calcium levels, and the intracellular calcium increase during efferocytosis was also observed using Fura2 (**Supplementary Fig. 8 in the revised manuscript**).

Fig. 3: The pyrazole compound used as SOCE channel inhibitor, BTP2, aka YM-58483, is not specific. It activates TRPM4 channels <https://www.ncbi.nlm.nih.gov/pubmed/16407466>, inhibits TRPC3 and TRPC5 channels <https://www.ncbi.nlm.nih.gov/pubmed/15647288> and binds to the actin reorganizing protein drebrin <https://www.ncbi.nlm.nih.gov/pubmed/19948240>. A more specific inhibitor such as GSK-5498A should be used to confirm the pharmacological findings.

As the reviewer suggested, we tested whether GSK-5498A could also inhibited efferocytosis. In the presence of GSK-5498A, efferocytosis by *WT* and *Crbn*^{-/-} BMDMs was decreased although a degree of inhibition by GSK-5498A was lower than that by YM-58483. In addition, efferocytosis by *Crbn*^{-/-} BMDMs was more severely affected by the inhibitor than that by *WT* BMDMs, resulting in comparable efferocytosis in *WT* and *Crbn*^{-/-} BMDMs in the presence of GSK-5498A (Supplementary Fig. 10c). Once again, these data suggest that the effect of *Crbn* on efferocytosis is caused though SOCE channels. We thank the reviewer for pointing out the specificity of the inhibitor and recommending a new experiment using GSK-5498A.

Figs 4-5: The study would benefit from the inclusion of quantitative evaluations of the western blots

As the reviewer suggested, we quantified the intensities of bands in the immunoblots of Fig.4a,b, Fig.4f, and Fig.6b in the original manuscript (Supplementary Fig. 11a-c, 13a in the revised manuscript).

Fig 6A. Crbn expression is said to reduce the engulfment on apoptotic cells, but no significance is indicated in the graph between these conditions.

First of all, we are sorry that we were not careful for the figure. Because we showed the effects of *Crbn* overexpression on efferocytosis in Fig. 1a in the original manuscript, we did not put the statistical significance between GFP and *Crbn* in Fig. 6a in the original manuscript although there was statistical significance between the samples.

In addition, the small effect of *Crbn* or *Crbn* and *Orai1* on efferocytosis in Fig 6a might be caused by the way for an efferocytosis assay. To evaluate the effect of a gene on efferocytosis, in some case we co-transfect phagocytes with a gene and GFP, incubate the phagocytes with TAMRA-stained apoptotic cells, and analyze the cells using flow cytometry. We consider GFP- and TAMRA-positive cells as phagocytes engulfing apoptotic cells. Thus, it is considered that GFP-positive phagocytes as phagocytes expressing the gene. Sometimes, although this experimental way is useful, it underestimates the effect of the gene on efferocytosis because GFP-positive phagocytes not always represent cells expressing the gene. Especially, when transfection efficiency is low, the possibility that GFP-positive cells express the gene is low. The effect of *Crbn* overexpression on efferocytosis in Fig. 6a was evaluated in this way. Thus, the small effect of *Crbn* on efferocytosis in Fig. 6a resulted from gating GFP-positive cells rather than *Crbn*-

positive cells and low transfection efficiency (Fig. 1a in the original manuscript shows the more apparent effect of Crbn than Fig. 6a due to high transfection efficiency).

Thus, we re-evaluate the effects of Crbn on efferocytosis by gating Crbn-positive cells. LR73 cells transfected with HA-Crbn were incubated with TAMRA-stained apoptotic cells, incubated with a FITC conjugated anti-HA antibody, and analyzed using flow cytometry. HA- and TAMRA-positive cells were considered as Crbn overexpressing phagocytes engulfing apoptotic cells. The effect of Crbn overexpression on efferocytosis was prominent, and Orai1 co-expression with Crbn apparently rescued the phenotype of efferocytosis by phagocytes overexpressing Crbn (Fig. 6a in the revised manuscript and please, also refer to the response to the 6th comment of reviewer #2).

Fig. 6b: The increased Orai1 abundance in macrophages after 30-60 minutes of phagocytosis might reflect the detection of channel protein coming from internalized thymocytes. T cells express high amounts of Orai1 channels, whose activity is required for their proliferation. This could be tested by using thymocytes expressing a tagged version of the Orai1 channel as phagocytic prey.

Thank you for pointing out the possibility that the increased Orai1 in BMDMs incubated with apoptotic thymocytes may come from internalized apoptotic thymocytes. Indeed, it is an important issue in the field to make it clear whether a protein of interest in phagocytes is contaminated with a target-derived protein. We are sorry that we did not address this point sufficiently in the original manuscript.

First of all, internalized apoptotic cells are rapidly degraded in phagocytes. Thus, it is generally believed in the field that contamination from internalized apoptotic cells is ignorable. Usually, however, the contamination results from unengulfed apoptotic cells because unengulfed apoptotic cells are not completely removed after washing. Due to this issue, we extensively wash phagocytes after incubation with apoptotic cells. Apoptotic cells remain rarely after the wash.

Second, the level of *Orai1* transcript in BMDMs is about 30 folds higher than that in thymocytes (BioGPS database, BMDMs versus thymocytes (CD4 and CD8 double positive thymocytes)). Thus, even though there is Orai1 contamination from unbound or internalized apoptotic thymocytes, its contribution to Orai1 levels in BMDMs might be negligible.

Third, we directly compared the level of Orai1 in BMDMs with that in apoptotic thymocytes used in Fig. 6b (up to 10 folds). We failed to detect Orai1 in apoptotic thymocytes. Even in apoptotic cells 10 times more than the apoptotic cells used in Fig. 6b, Orai1 could not be detected (Supplementary Fig. 13b in the revised manuscript). It seems that the relatively low transcript levels of *Orai1* in thymocytes and the

degradation of Orai1 during apoptosis make Orai1 undetectable in apoptotic thymocytes. Thus, the increased level of Orai1 in BMDMs during efferocytosis unlikely results from apoptotic thymocyte-derived Orai1. We hope that the experiment and comments relieve the reviewer's concern.

Fig. 6d: related to the point above, the decreased Crbn binding to Orai1-FLAG might reflect scavenging of Crbn by Orai1 from thymocytes.

We believe that the response to the above comment addresses this point. In addition, we performed new experiments to address a mechanism by which Crbn-Orai1 interaction is attenuated during phagocytosis of apoptotic cells. Neither altered subcellular localization nor modification of Crbn was provoked during efferocytosis. However, interaction between Orai1 and Stim1 was induced upon addition of apoptotic cells to phagocytes. In addition, overexpression of Stim1 weakened the interaction between Orai1 and Crbn, suggesting that Crbn competes with Stim1 for binding to Orai1 and that induced interaction between Orai1 and Stim1 during efferocytosis weakens interaction of Orai1 with Crbn (Fig. 6f-i in the revised manuscript and Supplementary Fig. 14, and please, also refer to the response to the 6th comment of reviewer #1).

The interaction between Orai1 and Crbn may be indirect and it would be interesting to see if this interaction is complexed with other proteins.

To address the point, we performed a yeast two-hybrid assay. Crbn orthologues are found in vertebrates and plants but have not been found in yeast, suggesting that a protein (if any) mediating interaction between Crbn and Orai1 unlikely exists in yeast. Therefore, interaction of Crbn with Orai1 in yeast indicates that the interaction could be direct. In addition, yeast is eukaryotic cells in which Crbn is translated more appropriately than in bacteria. Thus, using yeast system is advantageous to evaluate eukaryotic cell-derived protein-protein interaction. The fragments whose interaction was confirmed in mammalian cells were cloned into yeast expression vectors and performed a yeast two-hybrid assay. Yeast transformants expressing both Orai1^{N-term} and Crbn grew on the selective plate whereas transformants expressing only either Orai1^{N-term} or Crbn failed to grow on it (Fig. 4j in the revised manuscript). In addition, the interactions between the fragments of Crbn and Orai1 and between Crbn and ORAI1 β might be also further evidence supporting direct interaction between Crbn and Orai1. Thus,

the interactions between Crbn and Orai1 in yeasts and in mammalian cells strongly indicate that interaction between Crbn and Orai1 is direct.

Minor Comments:

In line 88-89 – ‘mental retardation’ is no longer used and was changed to ‘intellectual disability’ in 2013 (ref. https://journals.lww.com/co-psychiatry/FullText/2013/05000/New_terminology_for_mental_retardation_in_DSM_5.6.aspx)

We regret that we have not fully explored the field and thank the reviewer for correcting the words. As the reviewer suggested, we changed ‘mental retardation’ to ‘intellectual disability’ in the revised manuscript.

Line 166 ‘though’ should be ‘through’

We thank you for indicating the typo. We corrected the typo in the revised manuscript.

REVIEWER COMMENTS

Reviewer #2 (Remarks to the Author):

In this revised manuscript, Moon et al. have addressed most of my original critiques with convincing new experiments and a thoughtful discussion of the points I had raised. Overall the authors have done a good job responding to the critiques and the manuscript has improved quite a bit. Specifically: (i) My questions about the detection of Orai1 by Western blotting have been addressed well. (ii) The authors conducted a mutagenesis analysis of lysine residues in the N terminus of Orai1 and found that mutation of K89 reduces Orai1 ubiquitination, which is very interesting. Unfortunately they did not test the effects of this point mutation on efferocytosis, which means that we do not have direct evidence that Crbn-mediated ubiquitination of Orai1 is indeed responsible for reduced efferocytosis. This is a crucial experiment in my opinion. (iii) The authors convincingly show (and discuss) that inhibition of the lysosomal degradation pathway with bafilomycin A has no effects on Orai1 levels, whereas inhibition of the proteasome with MG132 does. (iv) The authors conducted new experiments in which they deleted Orai1 and Orai2 by RNAi to test the effects on efferocytosis, which confirmed their earlier conclusions based on the overexpression of Orai1 and Orai2. (v) The authors now demonstrate that Crbn is not required for the phagocytosis of beads and bacteria (consistent with published reports), which is in contrast to its role in efferocytosis and therefore an important distinction. (vi) In my last critique I had argued that "the authors need to show the significance of their findings in vivo, for instance in an autoimmune disease model for SLE or RA." The authors' argue "we evaluated the effect of Crbn on efferocytosis using two different approaches in vivo, clearance of apoptotic cells in the thymus and peritoneum", which confirms their in vitro findings. However, I do not find these data very convincing, especially the reduced thymic cellularity of dexamethasone-injected Crbn^{-/-} mice, which the authors argue is due to increased efferocytosis. The authors argue that "validating the roles of Crbn at a pathological condition could be performed as a separate study" and that "The current scope of our study is to delineate a molecular mechanism by which Crbn modulates clearance of apoptotic cells." I am willing to accept that and not insist on any new experiments. However, if the emphasis of this paper is on elucidating the molecular mechanisms by which Crbn regulates efferocytosis, then a better validation of the role of Orai1 ubiquitination by Crbn is required by testing the effects of the Orai1 K89A mutation on efferocytosis (see point ii of this critique).

Reviewer #3 (Remarks to the Author):

The authors have performed additional experiments to address the points raised in the first round of review. The new data show that Crbn interacts weakly with the short Orai isoform and does not impact its ubiquitination and that Orai1 levels are normal in CD4 T lymphocytes from Crbn null mice. This somewhat reduces the impact of the findings since the regulation of Orai1 stability by Crbn seems to be context-dependent and restricted to a specific ubiquitin ligase, but the new data confirm the initial findings that Crbn interacts with Orai1 N terminus, which are well documented, I have no further suggestions for changes.

Response to Referee #2

Reviewer #2 (Remarks to the Author):

In this revised manuscript, Moon et al. have addressed most of my original critiques with convincing new experiments and a thoughtful discussion of the points I had raised. Overall the authors have done a good job responding to the critiques and the manuscript has improved quite a bit. Specifically: (i) My questions about the detection of Orai1 by Western blotting have been addressed well. (ii) The authors conducted a mutagenesis analysis of lysine residues in the N terminus of Orai1 and found that mutation of K89 reduces Orai1 ubiquitination, which is very interesting. Unfortunately they did not test the effects of this point mutation on efferocytosis, which means that we do not have direct evidence that Crbn-mediated ubiquitination of Orai1 is indeed responsible for reduced efferocytosis. This is a crucial experiment in my opinion. (iii) The authors convincingly show (and discuss) that inhibition of the lysosomal degradation pathway with bafilomycin A has no effects on Orai1 levels, whereas inhibition of the proteasome with MG132 does. (iv) The authors conducted new experiments in which they deleted Orai1 and Orai2 by RNAi to test the effects on efferocytosis, which confirmed their earlier conclusions based on the overexpression of Orai1 and Orai2. (v) The authors now demonstrate that Crbn is not required for the phagocytosis of beads and bacteria (consistent with published reports), which is in contrast to its role in efferocytosis and therefore an important distinction. (vi) In my last critique I had argued that "the authors need to show the significance of their findings in vivo, for instance in an autoimmune disease model for SLE or RA." The authors' argue "we evaluated the effect of Crbn on efferocytosis using two different approaches in vivo, clearance of apoptotic cells in the thymus and peritoneum", which confirms their in vitro findings. However, I do not find these data very convincing, especially the reduced thymic cellularity of dexamethasone-injected Crbn^{-/-} mice, which the authors argue is due to increased efferocytosis. The authors argue that "validating the roles of Crbn at a pathological condition could be performed as a separate study" and that "The current scope of our study is to delineate a molecular mechanism by which Crbn modulates clearance of apoptotic cells." I am willing to accept that and not insist on any new experiments. However, if the emphasis of this paper is on elucidating the molecular mechanisms by which Crbn regulates efferocytosis, then a better validation of the role of Orai1 ubiquitination by Crbn is required by testing the effects of the Orai1 K89A mutation on efferocytosis (see point ii of this critique).

We appreciate the reviewer's careful point-by-point comments about the revised manuscript. In particular, we sincerely thank the reviewer for understanding and accepting our claims for *in vivo* studies using the disease model and for spending a lot of valuable time reviewing the manuscript. As the reviewer requested, we validated the effects of the degree of Orai1 ubiquitination on efferocytosis. Orai1 consistently increased engulfment of apoptotic cells as previously observed (Fig. 6a), and Orai1^{K89A} enhanced efferocytosis as well. Although the level of efferocytosis mediated by Orai1^{K89A} was slightly higher than that of efferocytosis mediated by Orai1, it was not statistically significant. However, the effect of Orai1^{K89A} on efferocytosis was different from that of Orai1 on efferocytosis when Crbn was co-expressed. Crbn suppressed efferocytosis mediated by Orai1^{K89A} less efficiently than by

Orai1, and the level of efferocytosis mediated by Orai1^{K89A} was significantly higher than that mediated by Orai1 in the presence of Crbn overexpression as measured by the percentage of phagocytes engulfing apoptotic cells (Supplementary Fig. 13a in the revised manuscript). Then, we tested whether the different effect of Orai1^{K89A} with Crbn on efferocytosis is due to the difference in the levels of Orai1 and Orai1^{K89A}. The expression level of Orai1^{K89A} was comparable to that of Orai1 in the absence of Crbn co-expression. Crbn co-expression drastically reduced the levels of both Orai1 and Orai1^{K89A}. However, Orai1 was affected more severely by Crbn co-expression than Orai1^{K89A}, and thus the expression level of Orai1^{K89A} than Orai1 was higher when Crbn was co-expressed (Supplementary Fig. 13b in the revised manuscript), which suggests that Orai1^{K89A} degrades more slowly than Orai1, resulting in the less reduced efferocytosis mediated by Orai1^{K89A} when Crbn is co-expressed. Taken together, these new data strengthen the molecular mechanism that Crbn modulates calcium flux through regulating the level of Orai1 during efferocytosis. We thank the reviewer for suggesting this critical experiment further clarifying the molecular mechanism.

Response to Referee #3

Reviewer #3 (Remarks to the Author):

The authors have performed additional experiments to address the points raised in the first round of review. The new data show that Crbn interacts weakly with the short Orai isoform and does not impact its ubiquitination and that Orai1 levels are normal in CD4 T lymphocytes from Crbn null mice. This somewhat reduces the impact of the findings since the regulation of Orai1 stability by Crbn seems to be context-dependent and restricted to a specific ubiquitin ligase, but the new data confirm the initial findings that Crbn interacts with Orai1 N terminus, which are well documented, I have no further suggestions for changes.

We are pleased that the reviewer accepted the revised manuscript. The critiques raised by the reviewer improved the rigor and clarity of the manuscript. Once again, we sincerely thank the reviewer for reviewing the manuscript and for spending his/her valuable time reviewing it.

REVIEWERS' COMMENTS

Reviewer #2 (Remarks to the Author):

The authors have conducted additional experiments shown in Supplemental Figure 13 to test the effects of the Orai1 K89A mutation on (a) Orai1 levels and (b) efferocytosis. These data show that overexpressed Crbn (a) cannot degrade Orai1 K89A as efficiently as it does wildtype Orai1, and (b) is less efficient in suppressing efferocytosis when coexpressed with Orai1 K89A compared to wildtype Orai1. These experiment add important evidence to support the idea that Crbn modulates efferocytosis through ubiquitination of Orai1. I am ready to recommend acceptance of the paper, but have a few suggestions (none of which require additional experiments): (1) consider including data from Suppl Figure 13 in Figure 6 (or another main figure). (2) Include significance levels in Suppl Figure 13 (are differences between Orai1 and Orai1 + Crbn significant? are differences between Orai1 K89A and Orai1 K89A + Crbn significant?). (3) Quantitate and average intensities of Orai1 bands from repeat Western blots and provide significance levels. (4) Mention in manuscript text that Orai1 K89A ONLY affects efferocytosis and Orai1 levels when Crbn is overexpressed. This is mentioned in the rebuttal letter, but not in the manuscript text.